# FreeFuse: Multi-Subject LoRA Fusion via Adaptive Token-Level Routing at Test Time

## Abstract

This paper proposes FreeFuse, a training-free framework for multi-subject text-to-image generation through automatic fusion of multiple subject LoRAs. In contrast to prior studies that focus on retraining LoRAs to alleviate feature conflicts, our analysis shows that spatially routing LoRA residuals to their intended semantic regions provides an effective mechanism for suppressing direct cross-region LoRA interference while preserving the base model's global contextual reasoning. Accordingly, we implement Adaptive Token-Level Routing during the inference phase. However, obtaining reliable routing regions remains challenging. Existing methods that rely on text-image latent association, such as raw cross-attention or concept-level similarity matching, often suffer from sparse activations, hole artifacts, and unstable localization when handling visually similar subjects, leading to incomplete or ambiguous subject masks. To address these issues, we introduce FreeFuseAttn, a mechanism that exploits the flow matching model's intrinsic semantic alignment to dynamically match subject-specific tokens to their corresponding spatial regions at early denoising timesteps, thereby bypassing the need for external segmentors. FreeFuse distinguishes itself through high practicality: it necessitates no additional training, model modifications, or user-defined spatial constraints. Users need only provide subject activation words to achieve seamless integration into standard workflows. Extensive experiments validate that FreeFuse outperforms existing approaches in both identity preservation and compositional fidelity. Our code is available at `https://anonymous.4open.science/r/FreeFuse_anno-FC99`.

## 1 Introduction

Large-scale text-to-image (T2I) models such as FLUX.1-dev (Labs, 2024; Cai et al., 2025) have demonstrated remarkable performance in general T2I tasks. To enhance their capability for personalized generation, Low-Rank Adaptation (LoRA) (Hu et al., 2022) has emerged as a preferred approach due to its precise fine-tuning quality and computational efficiency in both training and inference. LoRA also enables a simple way for multi-subject generation: Multiple subject LoRAs can be directly combined on the pretrained T2I models for multi-subject generation. However, this straightforward approach can lead to performance degradation, with the appearance of feature conflicts and deterioration (Shah et al., 2024; Kong et al., 2024; Kwon et al., 2024; Meral et al., 2024; Dalva et al., 2025; Po et al., 2024), making multi-subject LoRA fusion a challenging problem.

Prior works on multi-LoRA generation (Shah et al., 2024; Gu et al., 2023; Kong et al., 2024; Meral et al., 2024; Kwon et al., 2024) rely on designated techniques such as retraining, additional trainable parameters, external segmentation models or requiring users to provide template prompts or directly constrain the regions where LoRAs take effect, yet still struggle with multi-subject generation in complex scene. While recent approaches (Kong et al., 2024; Kwon et al., 2024; Dalva et al., 2025; Meral et al., 2024) mitigate conflicts via heavy inference interventions like noise or residual blending, these methods often compromise global coherence and inflate latency. We identify the root cause of feature conflict not as a generation artifact, but as the indiscriminate broadcasting of LoRA parameter updates ($\Delta\theta$). We demonstrate that routing additive LoRA residuals to their intended semantic regions effectively suppresses direct cross-region interference.

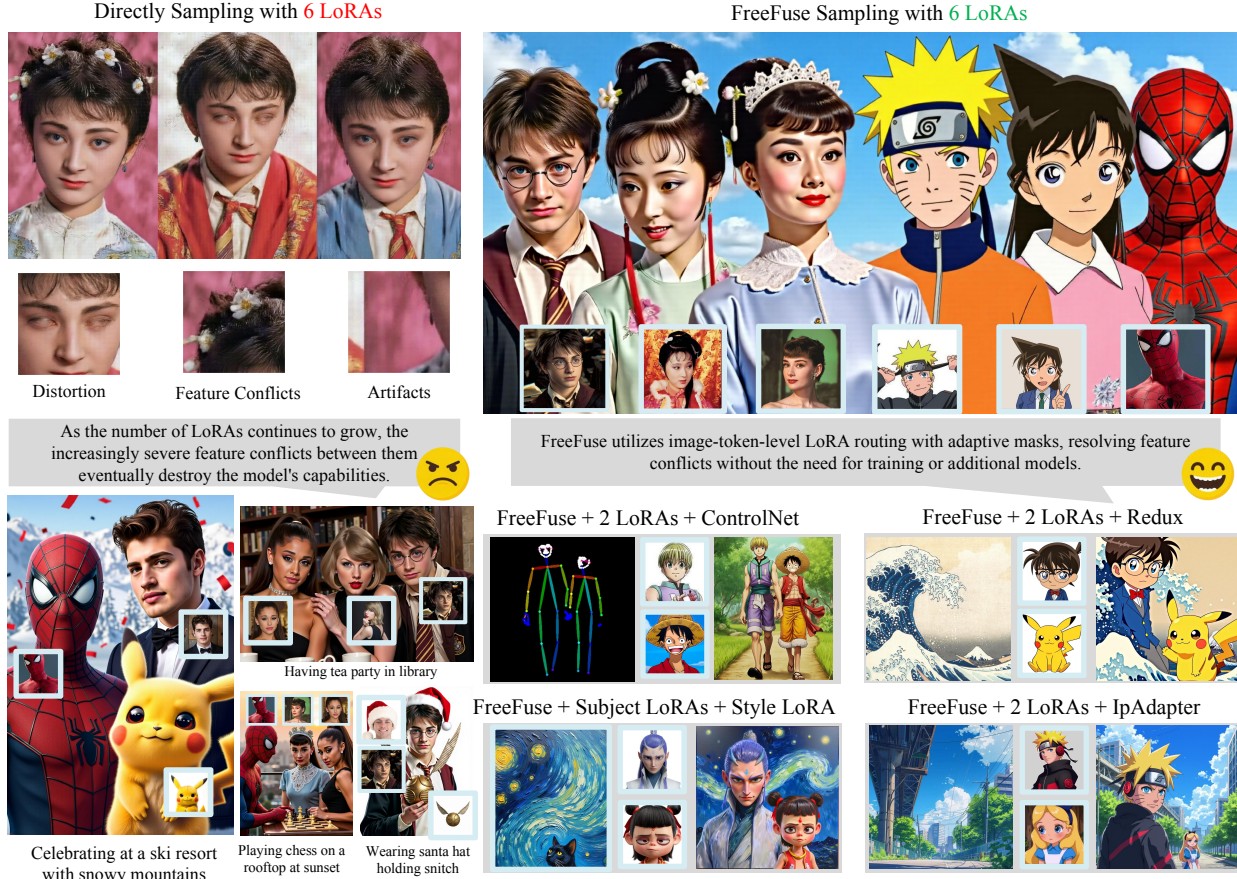

Figure 1: **Multi-subject-LoRA generation with FreeFuse**. We propose a robust, training-free framework that mitigates feature conflicts among multiple LoRAs by spatially constraining their influence. Our core mechanism, FreeFuseAttn, fuses semantic-driven cross-attention with cohesion-driven token similarity to generate contiguous subject masks, effectively reducing "hole" artifacts. The framework achieves high identity fidelity and is plug-and-play compatible with mainstream control modules without requiring auxiliary networks.

FreeFuse operates in two phases. In the first phase, corroborated by prior findings (Kwon et al., 2024; Helbling et al., 2025; Zhang et al., 2024; Choi et al., 2022) and our empirical observations, we introduce FreeFuseAttn, which capitalizes on the base model's latent segmentation capacity to construct an Image-Token-Level Router strictly during the early denoising stages. Simultaneously, we derive an Attention Bias from these high-fidelity segmentation cues to encourage robust alignment between spatial regions and subject semantics. In the second phase, the router orchestrates the multi-LoRA inference process by assigning each spatial token to a semantic subject/group and applying the adapters associated with that group. This supports both one-LoRA-per-subject and multi-adapter descriptions of a single subject, while still enforcing exclusivity between competing subject groups. Crucially, under the guidance of the Attention Bias, the model prioritizes subject-specific semantics, thereby significantly reducing concept bleeding.

In summary, our core contributions to the community include: (1) We provide a mechanism analysis demonstrating that spatially constraining additive LoRA residuals to target regions is an effective and empirically supported way to mitigate feature conflicts among multiple subjects. (2) FreeFuseAttn, which mitigates the 'hole' artifacts in latent space segmentation by fusing semantic-driven cross-attention with cohesion-driven token similarity. This approach ensures more complete and contiguous subject masks, outperforming cross-attention, ConceptAttention or SP-Attn methods in multi-object generation scenarios. (3) A robust, training-free multi-subject generation framework that fully harnesses the intrinsic capabilities of

the base model, eliminating the need for external segmentation modules or auxiliary networks. Due to its lightweight design, our approach offers seamless plug-and-play compatibility with mainstream control modules, such as spatial guidance tools (e.g., ControlNet), reference adapters (e.g., IP-Adapter, Redux), and aesthetic fine-tuners (e.g., Style LoRAs). (4) Extensive experiments confirm FreeFuse's superiority. Quantitative results establish a new state-of-the-art in identity preservation and compositional fidelity without compromising global image aesthetics.

## 2 Related Work

### 2.1 Text-to-Image Diffusion Model

In recent years, image generation models have advanced rapidly, evolving from early GAN-based models (Goodfellow et al., 2014; Arjovsky et al., 2017; Karras et al., 2019; 2020) to U-Net-based diffusion models (Ronneberger et al., 2015; Ho et al., 2020; Song et al., 2020; Rombach et al., 2022), and further to the widely adopted DiT-based diffusion models (Podell et al., 2023; Peebles & Xie, 2023; Esser et al., 2024; Labs, 2024; Cai et al., 2025; Wu et al., 2025a; Labs et al., 2025). With the continuous growth of model size, training scale, and architectural improvements, large-scale DiT-based models such as FLUX.1-dev (Labs, 2024) have become leaders among open-source models, while also driving research into customized generation, local editing, and style transfer.

### 2.2 Personalized Image Generation

Customized generation in diffusion models has been extensively studied. Textual inversion (Gal et al., 2022) methods encode rich semantic information into one or several text tokens through training. IP-Adapter (Ye et al., 2023),FLUX-Redux (Labs, 2024) and InstantID (Wang et al., 2024) instead train a generalizable module that directly takes one or more images and encodes their semantics into features aligned with the text or latent space. DreamBooth (Ruiz et al., 2023) introduces new concepts by fine-tuning diffusion network weights. With the wide adoption of LoRA (Hu et al., 2022) as an efficient fine-tuning method, fine-tuning open-source diffusion models with LoRA for customized generation has become a common choice among community users. Numerous works further improve LoRA or its training strategies, such as LyCORIS (Yeh et al., 2023), QLoRA (Dettmers et al., 2023), ED-LoRA (Gu et al., 2023), and SD-LoRA (Wu et al., 2025c), but LoRA itself remains the most widely used solution.

### 2.3 Multi Concept Generation

Early attempts such as ZipLoRA (Shah et al., 2024) and K-LoRA (Ouyang et al., 2025) fuse multiple LoRAs prior to inference. While successful in style transfer, these methods exhibit limited performance in preserving identities during multi-subject generation. Multi-LoRA (Zhong et al., 2024) introduces switch and composite strategies to mitigate conflicts; however, it struggles to delineate boundaries in complex multi-character scenarios. To enforce spatial isolation, methods like OMG (Kong et al., 2024), Concept Weaver (Kwon et al., 2024), and FlipConcept (Woo et al., 2025) employ auxiliary segmentation models combined with noise blending, yet these approaches often falter when delineating visually similar subjects (e.g., two men). While TokenVerse (Garibi et al., 2025) attempts to fuse concepts via Token Modulation, it remains suboptimal when handling semantic symmetry. Conversely, Mix-of-Show (Gu et al., 2023), Orthogonal Adaptation (Po et al., 2024), and LoRACLR (Simsar et al., 2025) require retraining LoRAs with manual spatial specifications; although they achieve high identity preservation, their rigid spatial constraints severely compromise compositional flexibility. Similarly, DreamRelation (Shi et al., 2025) offers fine-grained control but imposes a significant manual burden on the user. To circumvent these retraining costs and manual interventions, recent training-free approaches, including CLoRA (Meral et al., 2024), $MC^2$ (Jiang et al., 2025), and LoRAShop (Dalva et al., 2025), leverage cross-attention maps to derive concept masks. However, their performance degrades in complex scenes due to attentional leakage or maps deviating from semantic expectations. Furthermore, unified understanding-generation models like OmniGen (Xiao et al., 2025), Xverse (Chen et al., 2025), UNO (Wu et al., 2025b), and UMO (Cheng et al., 2025) explore multi-concept generation but are often restricted to a single reference image for each subject, leading to the loss of fine-

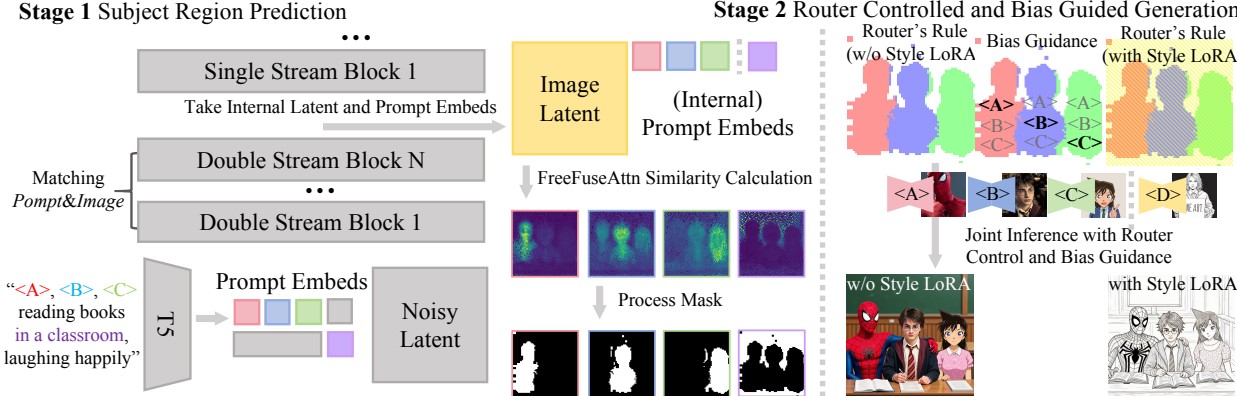

Figure 2: **The FreeFuse Pipeline.** In **Phase 1**, we employ FreeFuseAttn to extract robust subject masks, which are subsequently processed into a spatial Router and Attention Bias. In **Phase 2**, the Router enforces subject LoRA exclusivity per token to suppress direct cross-region LoRA interference, while the Bias mechanism actively reduces concept bleeding by ensuring precise semantic-spatial alignment.

grained identity details or layout overfitting. Distinct from these approaches, FreeFuse is positioned as a fully training-free and auxiliary-free LoRA fusion framework. By maximizing the model's intrinsic cross-modal alignment within the latent space, we synthesize high-quality multi-subject images with flexible, natural layouts, effectively liberating users from the burdens of LoRA retraining and manual spatial specification.

## 3 Methodology

In this section, we present FreeFuse, a comprehensive framework designed to resolve identity conflicts in multi-subject generation. We begin by demonstrating that naively aggregating LoRA outputs induces severe feature interference, and pointing out that spatially routing additive LoRA residuals to their semantic regions directly suppresses the major conflict path. Building on this insight, we systematically explore the intrinsic segmentation capabilities of Flow Matching models to localize subjects without external supervision, proposing **FreeFuseAttn** as a robust attention-based localization mechanism. Finally, we integrate these components into a two-stage pipeline: primarily, we leverage the model's intrinsic capabilities to predict precise subject masks; subsequently, we deploy a token-level Router and an attention Bias mechanism to strictly enforce local LoRA activation and suppress concept bleeding, enabling high-fidelity multi-subject synthesis.

### 3.1 Masking LoRA Outputs for Effective Subject Feature Preservation

Consider $S$ semantic subject groups $\{G_1, \ldots, G_S\}$, corresponding to spatial regions $\{R_1, \ldots, R_S\}$ in the latent space. Each group $G_s$ can be associated with one or more LoRA adapters $\mathcal{A}_s$. This group-level formulation supports both the common one-subject-one-LoRA case and cases where a single subject is jointly described by multiple adapters, such as an identity LoRA together with an attribute LoRA or a style LoRA. Directly merging all the LoRAs involves a naive summation of all LoRA outputs. This can introduce severe feature interference, as a token at position $p \in R_k$ is simultaneously perturbed by conflicting outputs from unrelated subject groups.

To enforce subject disentanglement, we utilize a spatial masking strategy. For a spatial token $p$, let $x_p$ denote the input hidden representation to the LoRA-augmented linear layer, $h_p$ denote the corresponding base-model output without LoRA residuals, and $h'_p$ denote the routed output after adding selected LoRA residuals. For any token position $p$, we retain only the contributions from adapters assigned to the corresponding semantic group:

$$h'_p = h_p + \sum_{s=1}^{S} \mathbb{I}(p \in R_s) \sum_{a \in \mathcal{A}_s} \Delta\theta_a(x_p), \tag{1}$$

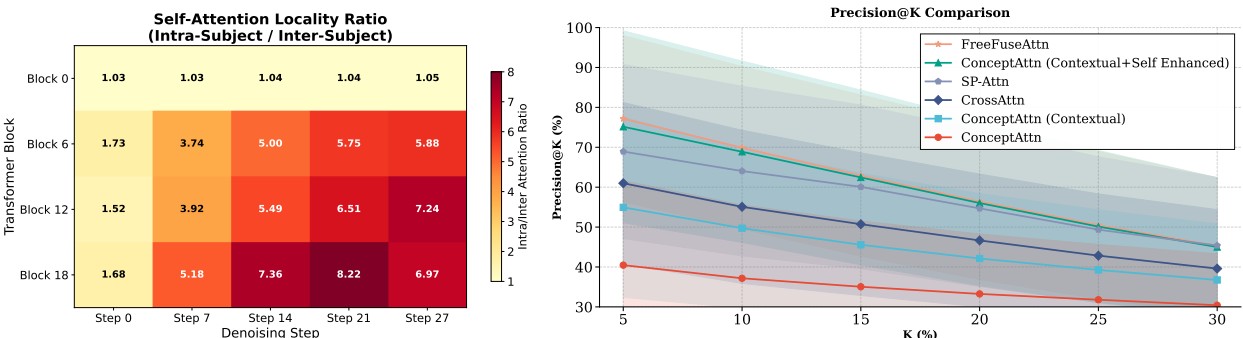

Figure 3: **Left:** Empirical analysis of attention locality on 100 samples using SAM3 masks. The heatmap reports the ratio of intra- to inter-subject attention weights. While early blocks aggregate global context (ratio $\approx 1$), deep layers exhibit strong spatial isolation (ratio $\gg 1$), validating our LoRA masking strategy. **Right:** We evaluate the spatial alignment of different mechanisms against SAM3-generated ground truth masks over 300 samples. **Precision@K** measures the percentage of the top $K\%$ activated tokens that correctly fall within the subject's region. FreeFuseAttn demonstrates superior localization accuracy.

where $\mathbb{I}(\cdot)$ is the indicator function and $\Delta\theta_a(x_p)$ denotes the additive residual produced by adapter $a$. This ensures that the LoRA feature update at region $R_k$ is governed only by the adapters assigned to group $G_k$, while LoRAs from competing groups are suppressed.

A potential concern is that LoRA features from disjoint regions $R_{j\neq k}$ might still propagate into $R_k$ through the global aggregation of the self-attention mechanism: For a self-attention layer with token features $X$, let $Q = XW_Q$, $K = XW_K$, and $V = XW_V$ denote the query, key, and value matrices. The attention matrix is $A = \text{Softmax}(QK^\top/\sqrt{d})$, where $A_{p,q}$ is the attention weight from query token $p$ to value token $q$:

$$\text{Attn}(Q, K, V)_p = \sum_q A_{p,q} V_q, \tag{2}$$

and $V_q$ is the value vector at token $q$. We do not claim that the base attention path prevents all cross-region information propagation; this global context exchange is useful for scene coherence. Instead, FreeFuse blocks the direct injection of subject-specific LoRA residuals into unrelated regions. For the remaining indirect propagation through base attention, we leverage the intrinsic ***spatial locality*** widely observed (Raghu et al., 2022; Caron et al., 2021; Helbling et al., 2025; Tian et al., 2024) in DiT models. As visualized in Fig. 3 Left, while the initial layers (e.g., Block 0) exhibit a ratio near 1.0, indicating a global context aggregation necessary for layout initialization, the ratio rises sharply in deeper semantic layers (reaching $> 7.0$ in Block 12-18). Crucially, this *diagonal dominance* becomes most pronounced during the semantic-forming denoising steps, where the attention map $A$ exhibits a strong diagonal dominance thus tokens in region $R_k$ predominantly attend to other tokens within the same region:

$$\sum_{q\in R_k} A_{p,q} \gg \sum_{q\notin R_k} A_{p,q}, \quad \forall p \in R_k. \tag{3}$$

We further verify this mechanism by measuring LoRA perturbation magnitudes across FLUX double-stream blocks, as shown in Fig. 4. The normalized effective LoRA weight perturbation is $1.80\times$ larger in mid/deep blocks than in early blocks, and the inference-time LoRA residual under the corresponding activation prompt is $2.67\times$ larger in mid/deep blocks. Therefore, although early blocks aggregate broader context, the LoRA-induced residuals being mixed there are empirically small; stronger LoRA effects emerge in mid-to-deep semantic blocks where attention locality is already much stronger. By applying spatial masks to the additive LoRA residual path, we effectively suppress direct cross-region LoRA interference while preserving the base model's global context propagation.

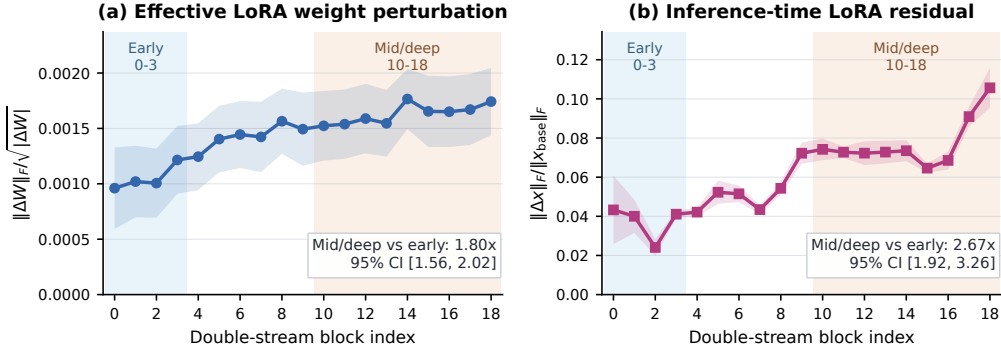

Figure 4: **LoRA perturbation magnitude across FLUX double-stream blocks.** Across all available LoRAs in the LoRAShop test set, early blocks carry substantially smaller LoRA perturbations than mid/deep blocks. Under the corresponding LoRA activation prompts, the inference-time LoRA residuals in mid/deep blocks are $2.67\times$ larger than in early blocks. Thus early global attention mainly mixes small LoRA residuals, while stronger LoRA effects emerge in more local semantic blocks.

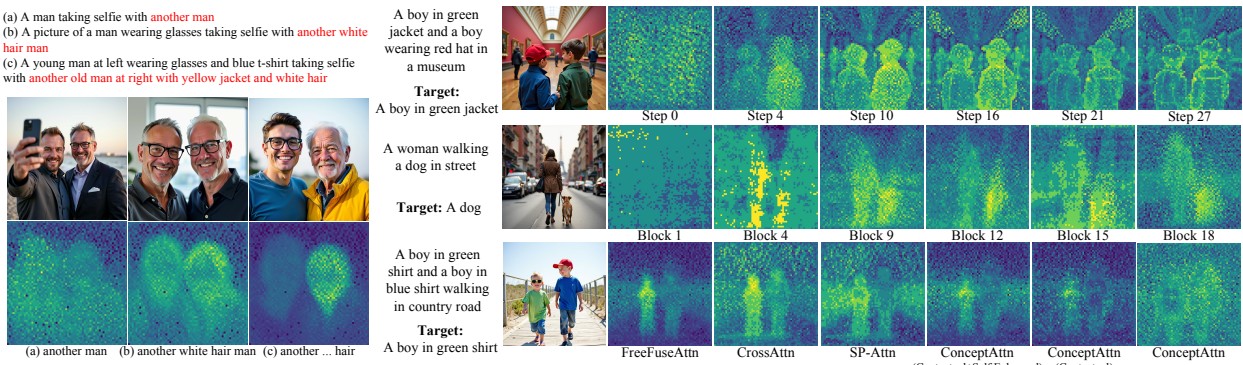

Figure 5: **Left**: High-quality descriptive prompts boost character distinguishability during generation and enhance similarity map quality. **Right**: Visualization of similarity maps across different dimensions. (a) Temporal Dynamics: Analyzing the cross attention heatmap in the denoising steps (top row) reveals that the alignment between text and image embeddings is most optimal during the early-to-mid stages. (b) Layer-wise Analysis: Within the Flux architecture (middle row), later blocks demonstrate better fusion of semantic and visual information. (c) Method Comparison: Compared to baseline methods (bottom row), our FreeFuse Attn exhibits the highest spatial discriminability, effectively disentangling symmetry concepts.

## 3.2 Exploring Intrinsic Segmentation Capabilities in Flow Models while Multi-Concept Generation

Prior literature (Baranchuk et al., 2021; Tang et al., 2023; Li et al., 2023; Xu et al., 2023; Tian et al., 2024; Helbling et al., 2025) has established that the latent representations of Diffusion and Flow Matching models exhibit superior semantic segmentation capabilities compared to discriminative baselines like CLIP or DINO. Whether achieved through post-hoc lightweight networks (Xu et al., 2023; Li et al., 2023; Baranchuk et al., 2021) or by exploiting self-attention and cross-attention clustering (Tang et al., 2023; Helbling et al., 2025; Tian et al., 2024), these findings corroborate the robust intrinsic segmentation potential of generative models. Building upon this premise, we investigate how to efficiently localize multiple subjects within the Flux (representing Flow Matching DiT) for multi-subject generation.

We first determine the optimal temporal window and architectural layer for mask extraction. Through empirical analysis on Flux.dev, we identify the early-to-mid denoising steps as the optimal temporal window. As shown in right line 1 of Fig. 5, we observe that at the initial timesteps, spatial structures remain nascent and indistinguishable from Gaussian noise; conversely, latents in later stages become dominated by high-

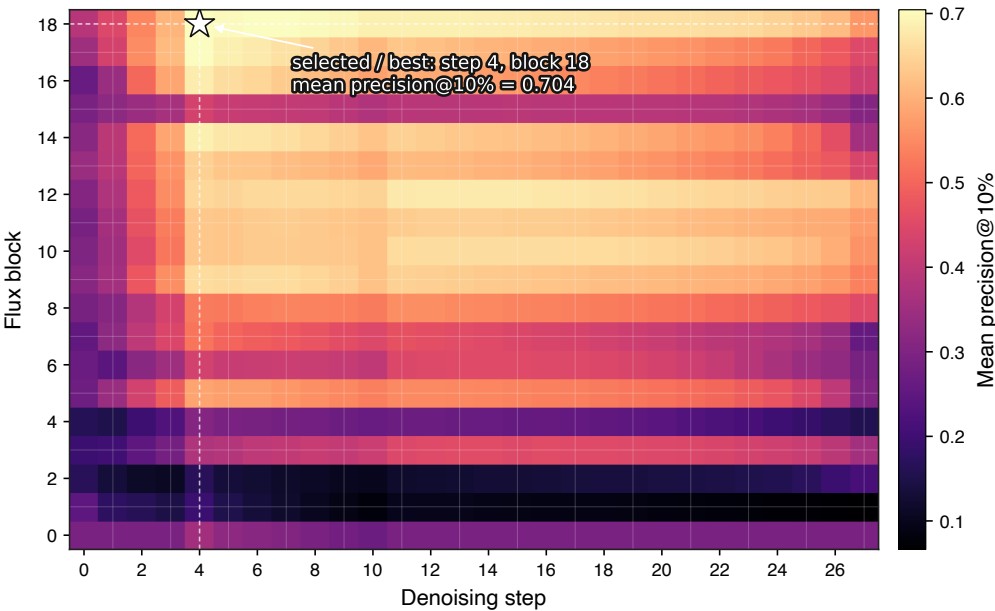

Figure 6: **Step/block sweep for FreeFuseAttn mask extraction.** We evaluate Precision@10%, defined as the percentage of the top 10% activated tokens that fall inside the corresponding subject region, over 300 generated samples. The selected cell, block 18 at step index 4, achieves the best mean precision of 0.704, supporting our use of the last double-stream block at the 5th denoising step.

frequency texture generation, showing a marked decoupling from semantic signals. This aligns with findings in (Choi et al., 2022; Tinaz et al., 2025; Qian et al., 2024; Lu et al., 2024), suggesting that the intermediate denoising phase represents the critical window where layout is established but not yet rigidified. We ultimately opted to extract the mask at the 5th step (step index 4 out of a total of 28 steps). We further evaluated semantic distinctiveness across different layers, pinpointing the output of the last Double Stream Block, `transformer_blocks.18`, as the optimal locus for semantic-image alignment (see right line 2 of Fig. 5). Fig. 6 further reports a quantitative step/block sweep under Precision@10%, confirming that block 18 at step index 4 yields the strongest localization accuracy. Comparative results for U-Net architectures (e.g., SDXL) are detailed in the Appendix. A.

With the optimal spatiotemporal locus established, we scrutinized mainstream mask extraction paradigms, specifically comparison against the widely adopted Cross-Attention (Meral et al., 2024; Dalva et al., 2025; Jiang et al., 2025) and the state-of-the-art ConceptAttn (Helbling et al., 2025). As depicted in the right line 3 of Fig. 5, we identify a critical limitation in ConceptAttn: its independent encoding of concepts leads to semantic collapse when handling visually similar subjects, as the embeddings lack contextual discrimination. Although implementing a contextual variant (encoding all concepts jointly) partially alleviates this, our empirical analysis reveals that standard Cross-Attention still retains the highest raw semantic fidelity for subject alignment. However, Cross-Attention maps are spatially fragmented, suffering from the 'hole phenomenon' identified in SPDiffusion (Zhang et al., 2024), where activations are sparse and non-target tokens exhibit significant noise. While SP-Attn (Zhang et al., 2024) attempts to mitigate this, it fails to achieve sufficient spatial cohesiveness in multi-subject optimization. Formally, let $Q \in \mathbb{R}^{N \times d}$ denote the spatial query features and $K_c \in \mathbb{R}^{L_c \times d}$ denote the key features corresponding to the tokens of concept $c$ (where $L_c$ is the token count). For clarity, we present the formulation for a single attention head; in practice, the scores are averaged across all heads. Unlike standard cross-attention which normalizes over the textual dimension, we aim to obtain a spatial probability distribution for each concept token. We compute the spatial similarity map

$\mathcal{A}_c \in \mathbb{R}^{N \times L_c}$ by applying Softmax along the spatial dimension $N$:

$$\mathcal{A}_c = \text{Softmax}_{\text{spatial}} \left( \frac{QK_c^\top}{\sqrt{d}} \right), \tag{4}$$

where the entry $\mathcal{A}_c^{(p,l)}$ represents the contribution of the $p$-th image token to the $l$-th concept token, such that $\sum_{p=1}^{N} \mathcal{A}_c^{(p,l)} = 1$. To derive the aggregate activation map $S_c \in \mathbb{R}^N$ for the entire concept, we average the spatial responses across all tokens associated with concept $c$:

$$S_c = \frac{1}{L_c} \sum_{l=1}^{L_c} \mathcal{A}_c[:, l]. \tag{5}$$

To identify distinct subject regions and suppress ambiguity, we compute the discriminative score $\hat{S}_c$. We enhance the signal by penalizing activations from competing concepts $j \neq c$:

$$\hat{S}_c = M \cdot S_c - \sum_{j \neq c} S_j, \tag{6}$$

where $M$ is the total number of concepts. We then identify the anchor set $\mathcal{P}_c = \text{TopK}(\hat{S}_c, k)$, pointing to the most representative spatial tokens. We use $k = \max(1, \lfloor 0.1N \rfloor)$, i.e., the top 10% spatial image tokens for each concept. For a $1024 \times 1024$ FLUX image, $N = (1024/16)^2 = 4096$, so $k = 409$ anchors are selected.

Finally, to generate a spatially cohesive mask $\mathcal{M}_c$ and mitigate the sparsity of raw cross-attention, we propagate semantic information from these anchors via latent similarity. Let $Z \in \mathbb{R}^{N \times d}$ denote the image-token feature matrix from the selected FLUX block, and let $Z_p \in \mathbb{R}^d$ denote the feature vector at anchor position $p$:

$$\mathcal{M}_c = \sigma \left( \frac{1}{\tau |\mathcal{P}_c|} \sum_{p \in \mathcal{P}_c} Z Z_p^\top \right). \tag{7}$$

Here, $Z Z_p^\top \in \mathbb{R}^N$ is a dense similarity map over image tokens, $\sigma(\cdot)$ denotes a spatial normalization function (e.g, min-max scaling), and we set the temperature to $\tau = 4000$. The resulting $\mathcal{M}_c \in \mathbb{R}^N$ is reshaped to the latent grid $H/16 \times W/16$. This effectively reconstructs the dense subject shape from sparse anchor cues. We further conducted a systematic evaluation on 300 generated samples involving dual-subject interactions. We utilized the Segment Anything Model 3 (SAM3) (Carion et al., 2025) to generate ground-truth segmentation masks for each subject. We then computed the Precision@K metric, defined as the proportion of tokens falling within the ground-truth mask among the top $K\%$ of tokens with the highest similarity scores. As illustrated in Fig. 3 Right, FreeFuseAttn consistently outperforms baseline methods across all $K$ thresholds. Notably, while Cross-Attention suffers from low precision due to its sparse activation nature, and standard ConceptAttn collapses due to semantic ambiguity, our method achieves the highest alignment with the semantic ground truth.

Additionally, empirical observations presented in Fig. 5 Left suggest that the efficacy of this reconstruction is positively correlated with the granularity of the textual descriptions. We find that supplementing subject prompts with distinctive attributes effectively mitigates the issue of map entanglement caused by semantic symmetry. This phenomenon aligns with intuition since richer semantic cues likely increase the orthogonality between concept embeddings in the latent space, thereby facilitating the identification of distinct anchors and yielding sharper similarity maps.

### 3.3 Router Controlled and Bias-Guided Generation

To preclude the mutual interference inherent in multi-LoRA activation, we perform an initial inference pass with all adapters deactivated. Leveraging the optimal spatiotemporal window and block identified in Sec. 3.2, we extract similarity maps for each subject. Unlike (Luo et al., 2024; Epstein et al., 2023; Dalva et al., 2025) which relies on rigid thresholding for foreground separation, we explicitly utilize a background

prompt to construct a competitive background similarity map, thereby significantly enhancing the precision of foreground-background delineation.

We subsequently refine these raw maps into robust binary masks using morphological opening and closing operations to ensure topological coherence. To resolve ambiguity in regions where subjects compete, we employ a contention resolution strategy incorporating territorial voting and gravity aggregation (refer to Appendix. B for full formulation).

With high-fidelity masks $\mathcal{M}$ established, we introduce the **Multi-LoRA Router Controlled and Bias Guided Generation** framework illustrated in Fig. 2 stage 2. While prior approaches(Meral et al., 2024; Sueyoshi & Matsubara, 2024; Rassin et al., 2023) attempt to mitigate the "concept bleeding" phenomenon via computationally expensive test-time optimization, our method directly leverages $\mathcal{M}$ to construct a spatial attention bias matrix. This matrix modulates the attention mechanism by encouraging image tokens to attend exclusively to their corresponding subject prompts while suppressing interactions with irrelevant subject descriptions. Concurrently, the Router enforces spatial locality by applying the adapter set $\mathcal{A}_s$ only to tokens inside the corresponding region $\mathcal{M}_s$. We define whole-image stylistic adjustment as the case where the user explicitly attaches a style LoRA to stylize the generated image rather than to modify a single localized identity or attribute, such as a Van Gogh painting style LoRA or a Miyazaki-style animation LoRA. Such a style adapter is handled by the same group-level rule: it is included in every foreground adapter set that should receive the style, i.e., $a_{\text{style}} \in \mathcal{A}_s$ for all relevant $s$; when the background should share the same style, we assign the style adapter to the background or full-canvas region as well.

**Implementation scope.** For FLUX.1-dev, routing is applied to all LoRA-bearing transformer blocks: the 19 double-stream blocks `transformer_blocks.0-18` and the 38 single-stream blocks `single_transformer_blocks.0-37`. In double-stream blocks, the Router gates image-token LoRA residuals in `to_q`, `to_k`, `to_v`, `to_out`, and the image feed-forward branch; subject text-token residuals are isolated in `add_q_proj`, `add_k_proj`, `add_v_proj`, `to_add_out`, and `ff_context`. In single-stream blocks, routing is applied to the image-token LoRA residuals in `to_q`, `to_k`, `to_v`, `proj_mlp`, and `proj_out`. During Phase 1, all adapters are disabled and masks are extracted from block 18 during the first $K = 5$ of $T_{\text{denoise}} = 28$ steps. During Phase 2, we restart from the same initial latent and apply routed LoRA residuals at every denoising step. Importantly, routing gates only the additive LoRA residuals; the frozen base-model attention remains active, preserving global scene composition.

## 4 Experiments

**Experimental Setup.** To ensure a rigorous evaluation, we benchmark FreeFuse against five methods representing three distinct paradigms. (1) LoRA-based Fusion: We select LoRAShop (Dalva et al., 2025) as the primary baseline. Crucially, we exclude older U-Net based methods to mitigate confounding variables arising from base model discrepancies, ensuring performance gains are attributable to our fusion algorithm rather than the Flux transformer's capabilities. (2) Modular Adapter Mechanisms: we focus on Multi-IP-Adapter and Multi-Redux as representative baselines. These exemplify the widely adopted "plug-and-play" paradigm, where lightweight modules inject visual conditions without altering base model weights. (3) Unified Native Models: We include OmniGen (Xiao et al., 2025) and UMO (Cheng et al., 2025)(UNO Based Version) to evaluate against emerging architectures designed with intrinsic multi-modal understanding.

### 4.1 Quantitative Results

To rigorously evaluate the versatility of FreeFuse across diverse semantic domains, we curated a comprehensive multi-subject benchmark consisting of 15 distinct LoRAs. The subject set is strategically categorized to test robustness against domain shifts and varying spatial granularities: it includes 10 Character LoRAs spanning photorealistic humans, 2D anime figures, 3D avatars, and anthropomorphic entities, alongside 5 Object LoRAs ranging from large-scale vehicles to deformable apparel and intricate accessories.

We push the evaluation complexity significantly beyond prior works(Kong et al., 2024; Dalva et al., 2025; Meral et al., 2024), which typically limited assessment to dual-subject scenarios. Our protocol involves two

Table 1: Compared to existing methods, our approach demonstrates a significant advantage in Face Similarity and improvements in character and object consistency, while remaining on par with state-of-the-art models in instruction following and aesthetic evaluations.

| Methods | Reference Similarity Metrics | | | | | | | | Face Similarity Metrics | | Prompt Following and Aesthetic Evaluation | | | User Study | |
|---|---|---|---|---|---|---|---|---|---|---|---|---|---|---|---|
| | Character Similarity | | | | Object Similarity | | | | | | | | | | |
| | DINOv2↑ | DINOv3↑ | Dreamsim↓ | Clip-I↑ | DINOv2↑ | DINOv3↑ | Dreamsim↓ | Clip-I↑ | ArcFace↑ | LVFace↑ | Clip-T↑ | HPSv2↑ | HPSv3↑ | Q1↓ | Q2↓ |
| OmniGen | 0.4699 | 0.5177 | 0.4233 | 0.6367 | 0.4662 | 0.5168 | 0.5048 | 0.6226 | 0.2990 | 0.1661 | 0.2088 | 0.2484 | 5.588 | 4.9 | 4.5 |
| UMO | 0.4378 | 0.4498 | 0.4710 | 0.5919 | 0.6535 | 0.6884 | 0.3635 | 0.7424 | 0.3180 | 0.1661 | 0.2277 | 0.2629 | 8.756 | 2.9 | 3.5 |
| Multi-Redux | 0.4433 | 0.4622 | 0.5363 | 0.5306 | 0.2456 | 0.2789 | 0.7255 | 0.3486 | 0.1475 | 0.0361 | 0.1225 | 0.2341 | 1.062 | 6.0 | 5.9 |
| Multi-IP-Adapter | 0.4795 | 0.4918 | 0.4707 | 0.6028 | 0.4790 | 0.4970 | 0.5304 | 0.6424 | 0.1840 | 0.0710 | 0.2911 | 0.2831 | 7.197 | 3.9 | 3.4 |
| LoRAShop | 0.4597 | 0.4974 | 0.4209 | 0.6567 | 0.6269 | 0.6394 | 0.4324 | 0.7129 | 0.3886 | 0.2350 | 0.2980 | 0.2880 | 8.494 | 2.2 | 2.7 |
| **Ours** | **0.4988** | **0.5235** | **0.3753** | **0.6764** | 0.6393 | 0.6804 | **0.3516** | **0.7499** | 0.4275 | 0.2534 | 0.2766 | 0.2857 | 8.279 | **1.1** | **1.1** |

Table 2: Ablation study results indicating the effectiveness of each component.

| Methods | Reference Similarity Metrics | | | | | | | | Face Similarity Metrics | | Prompt Following and Aesthetic Evaluation | | |
|---|---|---|---|---|---|---|---|---|---|---|---|---|---|
| | Character Similarity | | | | Object Similarity | | | | | | | | |
| | DINOv2↑ | DINOv3↑ | Dreamsim↓ | Clip-I↑ | DINOv2↑ | DINOv3↑ | Dreamsim↓ | Clip-I↑ | ArcFace↑ | LVFace↑ | Clip-T↑ | HPSv2↑ | HPSv3↑ |
| Ours (Cross-Attn) | 0.4271 | 0.4378 | 0.4838 | 0.5919 | 0.6405 | 0.6832 | 0.3867 | 0.7378 | 0.3119 | 0.1975 | 0.2537 | 0.2762 | 6.704 |
| Ours (w/o Postprocessing) | 0.4377 | 0.4733 | 0.4303 | 0.6323 | 0.6055 | 0.6570 | 0.4052 | 0.7392 | 0.3259 | 0.1868 | 0.2868 | 0.2846 | 7.727 |
| Ours (w/o Attn bias) | 0.4234 | 0.4691 | 0.4186 | 0.6448 | **0.6589** | 0.6939 | 0.3522 | **0.7635** | 0.3491 | 0.2027 | 0.2882 | 0.2865 | 7.971 |
| Ours (w/o Router; bias only) | 0.3759 | 0.4138 | 0.5065 | 0.5855 | 0.6495 | **0.7249** | 0.6546 | 0.7477 | 0.2669 | 0.1523 | 0.2678 | 0.2807 | 6.913 |
| **Ours(Full)** | **0.4988** | **0.5235** | **0.3753** | **0.6764** | 0.6393 | 0.6804 | **0.3516** | 0.7499 | **0.4275** | **0.2534** | 0.2766 | 0.2857 | **8.279** |

distinct stress tests: (1) Multi-Character Interaction: We designed 20 distinct scenes with diverse backgrounds, each necessitating the simultaneous inference of 4 concurrent LoRAs, thereby imposing a high load on the fusion mechanism. (2) Character-Object Interaction: We synthesized composite prompts by conjoining character descriptions with specific object-interaction templates. In total, 470 images were generated for each method to ensure statistical significance.

We assess performance using a multi-faceted metric suite that evaluates the generated results from three complementary perspectives: reference similarity between the generated images and the original character images, face similarity between each generated character and its corresponding ground-truth identity, and prompt following and aesthetic with respect to the given prompts. This allows us to comprehensively measure both character consistency and prompt-aligned generation quality. Specifically, we employ DINOv2 (Oquab et al., 2023) and DINOv3 (Siméoni et al., 2025) to measure coarse-grained visual fidelity, while CLIP-I (Radford et al., 2021) and DreamSim (Fu et al., 2023) capture high-level semantic consistency. For stringent identity verification of human subjects, we incorporate ArcFace (Deng et al., 2019) and LVFace (You et al., 2025). Global instruction adherence is quantified via CLIP-T. For aesthetic quality, the evaluation is conducted using HPSv2 (Wu et al., 2023) and HPSv3 (Ma et al., 2025). To validate perceptual alignment, we further conducted a user preference study (details in Appendix. G).

As evidenced in Tab. 1, FreeFuse establishes a new state-of-the-art in identity preservation. Our method demonstrates a commanding lead across both visual similarity metrics (DINOv2, DINOv3, Clip-I, Dreamsim) and fine-grained facial recognition benchmarks (ArcFace/LVFace). Furthermore, FreeFuse maintains superior object fidelity without compromising global image quality, achieving aesthetic scores (HPS) comparable to leading baselines while securing the top rank in human evaluations.

## 4.2 Qualitative Results

Fig. 7 presents a visual comparison against baseline methods. Observe that competing approaches frequently suffer from severe feature leakage and identity blending, where attributes of one subject (e.g., skin tone, facial structure) erroneously bleed into another. In contrast, FreeFuse enforces semantic isolation, preserving the distinct identity of each subject even in spatially proximate interactions. To further validate versatility, Fig. 8 displays a generation matrix involving 25 diverse characters and objects. Ranging from photorealistic humans to stylized anime figures and non-human entities, our method consistently synthesizes high-fidelity multi-subject interactions with coherent spatial layouts, demonstrating robust generalization across a broad semantic spectrum.

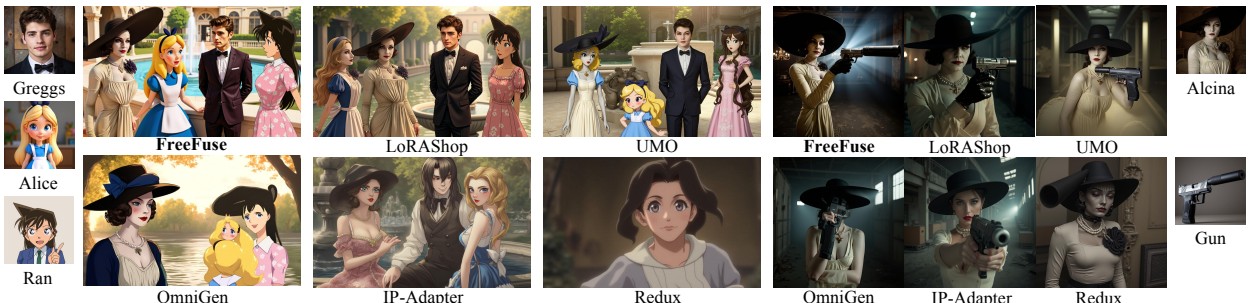

Figure 7: Qualitative Comparison. Prompt 1: *<A>, , <C>, and <D> are having a friendly conversation by the fountain pool*; Prompt 2: *<A> is aiming a <Gun>*.

Table 3: Inference latency comparison (seconds) at $1024 \times 1024$ resolution on a single L20 GPU. FreeFuse adds a mask-extraction pass, but remains substantially faster than LoRAShop while avoiding the severe conflicts of naive multi-LoRA sampling.

| Method | 2 LoRAs | 3 LoRAs | 4 LoRAs | 5 LoRAs | 6 LoRAs |
|---|---|---|---|---|---|
| Naive Flux | 21.95 | 24.60 | 27.11 | 29.64 | 32.08 |
| LoRAShop (Dalva et al., 2025) | 64.78 | 84.36 | 103.60 | 122.96 | 142.27 |
| **FreeFuse** | **36.08** | **42.96** | **50.29** | **58.63** | **67.96** |
| **Speedup vs. LoRAShop** | **1.80×** | **1.96×** | **2.06×** | **2.10×** | **2.09×** |

### 4.3 Ablation Study

To isolate the contribution of individual components, we conducted a systematic decomposition of the FreeFuse framework, with results summarized in Tab. 2. First, replacing our FreeFuseAttn with standard Cross-Attention yields a marked degradation in identity fidelity, confirming that raw attention maps are too sparse to support robust localization. Second, omitting the post-processing stage (morphological filtering and contention resolution) results in noisy masks; while general structure is maintained, the precision of boundary delineation suffers, leading to increased artifacts in complex interactions. Third, ablating the Bias-Guided Generation mechanism weakens semantic-spatial alignment and allows feature leakage to reappear. Finally, removing the Router while retaining only the attention bias produces a bias-only variant in which LoRA residuals are no longer spatially gated. Its substantial drop on character similarity and face identity metrics (e.g., ArcFace 0.2669 vs. 0.4275 for the full method) directly isolates the contribution of token-level LoRA routing.

**Discussion.** A salient advantage of FreeFuse lies in its seamless modularity. By eschewing external segmentors and invasive weight updates, our framework preserves the original manifold of the base model. While primarily demonstrated on Flow Matching transformers, the core mechanism of FreeFuse is theoretically architecture-agnostic and generalizes to other backbones (e.g., SDXL), as detailed in Appendix.A. This intrinsic design renders FreeFuse naturally compatible with downstream adapters: As illustrated in Fig. 1, it functions orthogonally to spatial guidance modules like ControlNet and reference encoders such as IP-Adapter or Redux. Consequently, FreeFuse not only resolves multi-subject conflicts but also unlocks fine-grained structural control and style transfer within complex compositional scenarios, expanding the applicability of pipelines.

## 5 Limitations and Future Work

Our framework involves a two-stage pipeline, mask extraction followed by router-guided inference, which inevitably introduces computational overhead. However, as shown in Tab. 3 and detailed in Appendix H,

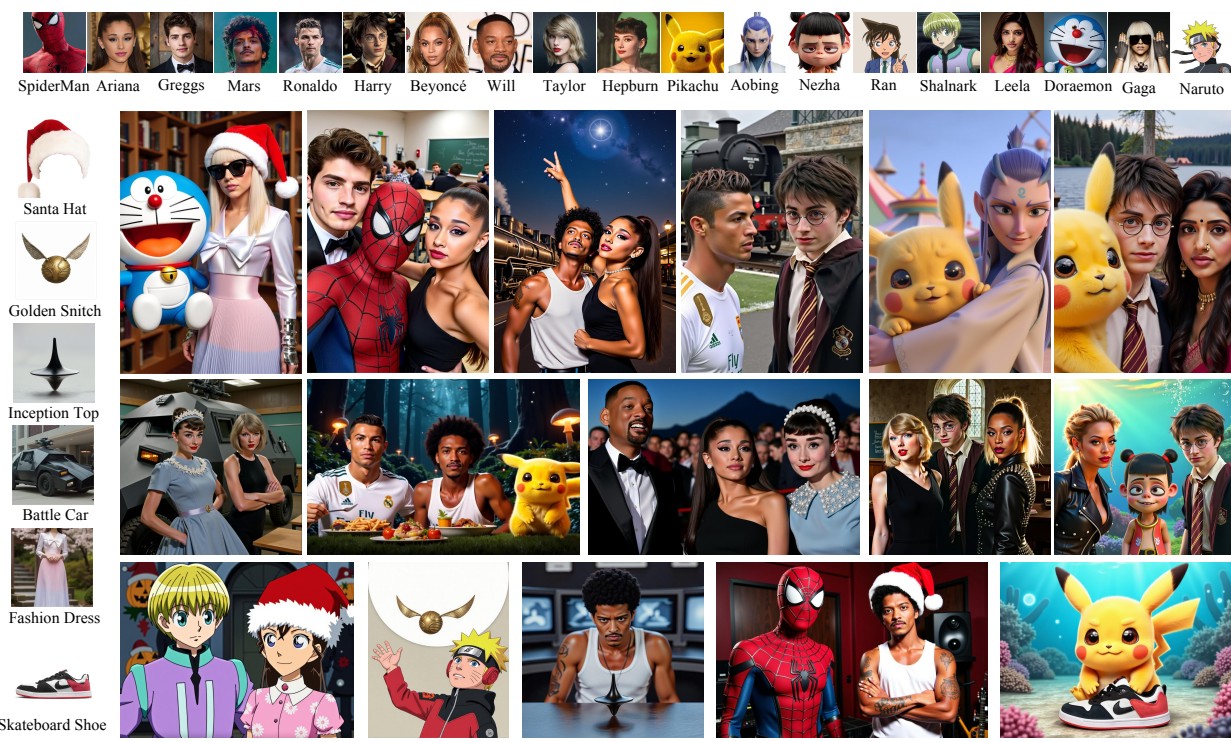

Figure 8: Additional visualization results. Our method achieves high-quality joint image generation across diverse categories, including real-world humans, anime/3D characters, non-human entities, garments, objects, and vehicles.

empirical benchmarks demonstrate that FreeFuse scales far more efficiently than existing training-free alternatives based on Flux (e.g., LoRAShop Dalva et al. (2025)). Future work will extend this mechanism to video generation, investigating how temporal attention dynamics can be leveraged to maintain multi-subject consistency across frames.

# 6 Conclusion

We present FreeFuse, a training-free framework for multi-subject LoRA fusion in text-to-image generation. FreeFuse mitigates feature conflicts by spatially routing each subject LoRA to its corresponding semantic region, without requiring LoRA retraining, auxiliary segmentation models, or user-specified masks. To obtain reliable routing regions, FreeFuseAttn leverages the intrinsic semantic alignment of diffusion transformers, combining cross-attention anchors with latent similarity propagation to produce dense and discriminative subject masks that alleviate sparse activations and hole artifacts. Based on these masks, adaptive token-level routing and bias-guided attention enforce local LoRA activation and suppress identity leakage during inference. Extensive experiments demonstrate that FreeFuse improves identity preservation, object consistency, and compositional fidelity while maintaining competitive prompt following and visual quality.

**Broader Impact Statement**

This research explores the technical possibilities of text-to-image compositional synthesis. We recognize the societal implications inherent in personalized generation technologies. The capability to integrate specific subject LoRAs carries the risk of misuse, including the creation of non-consensual content or misleading depictions (deepfakes). We strictly oppose the application of our method for generating NSFW content, harassment, or disinformation. We emphasize the importance of adhering to the safety guidelines and terms of use of the underlying foundational models.

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

# A Generalizability to U-Net Architectures (SDXL)

While our primary investigation focuses on the Flow Matching architecture (specifically FLUX.1-dev Labs (2024)) due to its superior prompt adherence and image quality, the core mechanism of FreeFuse, specifically the *FreeFuseAttn* for mask extraction, the *Token-Level Routing* for inference control and the *Bias-Guided Generation*, is theoretically architecture-agnostic. In this section, we validate the extensibility of our framework by applying it to SDXL Podell et al. (2023), a representative U-Net-based diffusion model.

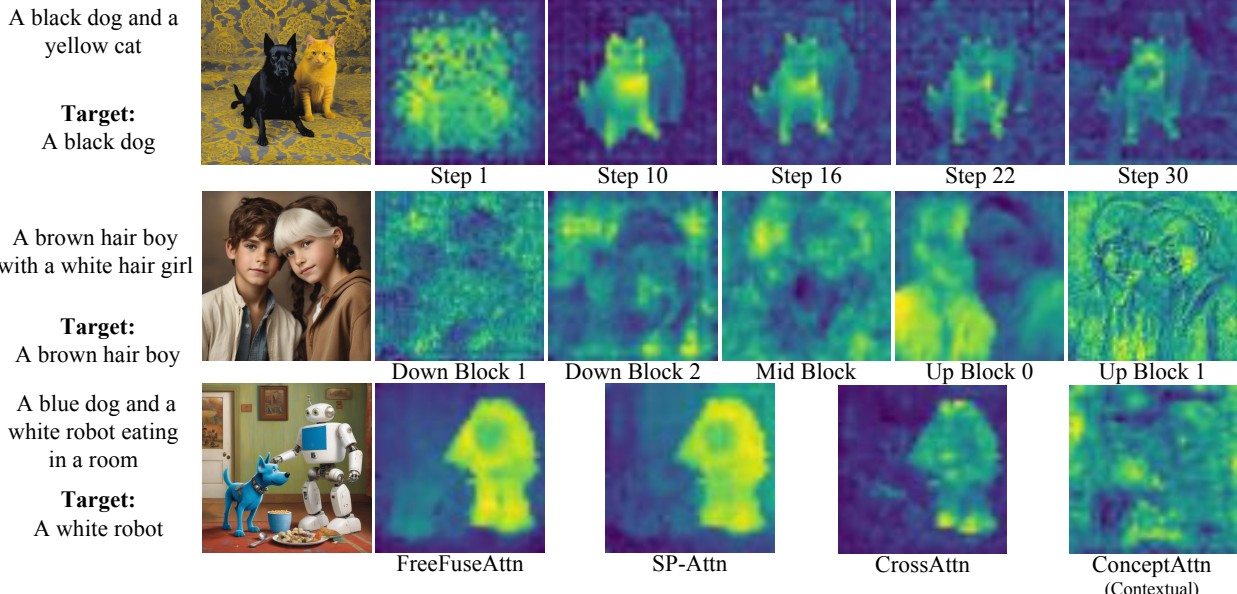

Figure 9: Visualization of similarity maps across different dimensions. (a) Temporal Dynamics: Analyzing the cross attention heatmap in the denoising steps (top row) reveals that the alignment between text and image embeddings is most optimal during the early-to-mid stages. (b) Layer-wise Analysis: Within the SDXL architecture (middle row), up block 0 demonstrate better fusion of semantic and visual information. (c) Method Comparison: Compared to baseline methods (bottom row), our FreeFuse Attn exhibits the highest spatial discriminability, effectively disentangling symmetry concepts.

## A.1 Implementation Details on SDXL

Unlike the unified DiT blocks in Flux, SDXL employs a separation of cross-attention and self-attention within the U-Net backbone. To adapt FreeFuse to this architecture, we implemented the following adjustments:

- **Layer Selection:** We extract attention maps from the mid-level cross-attention layers of the U-Net decoder (specifically, `up_blocks.0.attentions.0.transformer_blocks.3.attn2`). Empirical analysis suggests these layers contain the most semantic-rich spatial layout information necessary for mask generation.

- **Routing Application:** The token-level routing is applied to the corresponding self-attention layers, enforcing the exclusivity constraint defined in the main paper. This gates direct LoRA residual

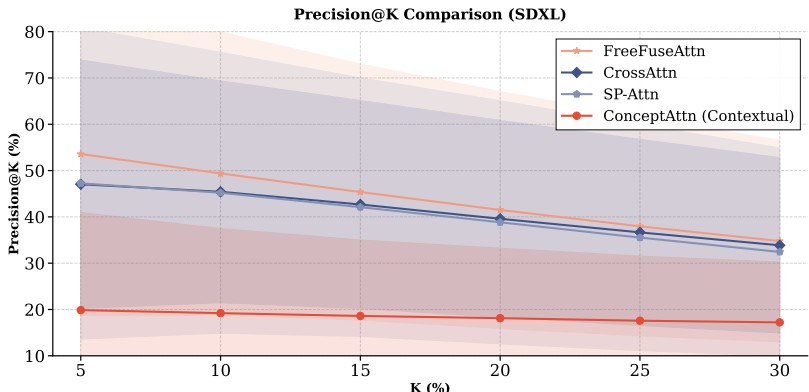

Figure 10: On SDXL, we evaluate the spatial alignment of different mechanisms against SAM3-generated ground truth masks over 300 samples. **Precision@K** measures the percentage of the top $K\%$ activated tokens that correctly fall within the subject's region. FreeFuseAttn demonstrates superior localization accuracy.

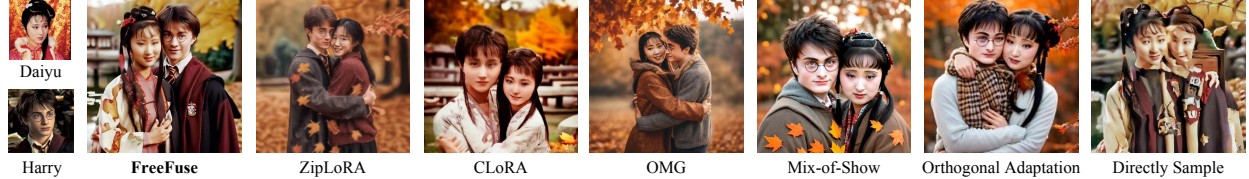

Figure 11: Qualitative results of FreeFuse applied to SDXL. Prompt: *harry_potter and daiyu_lin, both faces close together, autumn leaves blurred in the background*

    injection across conflicting subject regions while preserving the base model's pretrained attention pathway.

- **Bias-Guided Generation:** We extend the attention bias mechanism to the U-Net's cross-attention layers. By injecting a mask-derived bias matrix into the cross-attention scores, we actively suppress the attention weights of irrelevant subject prompts outside their designated regions (as defined by the masks extracted in the first stage), thereby minimizing semantic leakage.

These choices were determined through the same experimental procedures as those described in the main text; please refer to Fig. 9 and Fig. 10.

## A.2 Qualitative Analysis

Fig. 11 presents the qualitative results of FreeFuse applied to SDXL. The results demonstrate that our method performs well on SDXL, generating high-quality images without requiring any control signals other than text prompts. We further present the generation results on Illustrious-XL, currently the most popular SDXL-derived model on Civitai in Fig. 12.

**Note on Performance:** *While FreeFuse effectively mitigates concept bleeding in SDXL, we observe that the overall prompt adherence and spatial complexity handling of the base SDXL model are naturally lower than that of Flux. Consequently, our main quantitative benchmarks in Section 4 focus on the Flux architecture to decouple the fusion algorithm's performance from the base model's capabilities.*

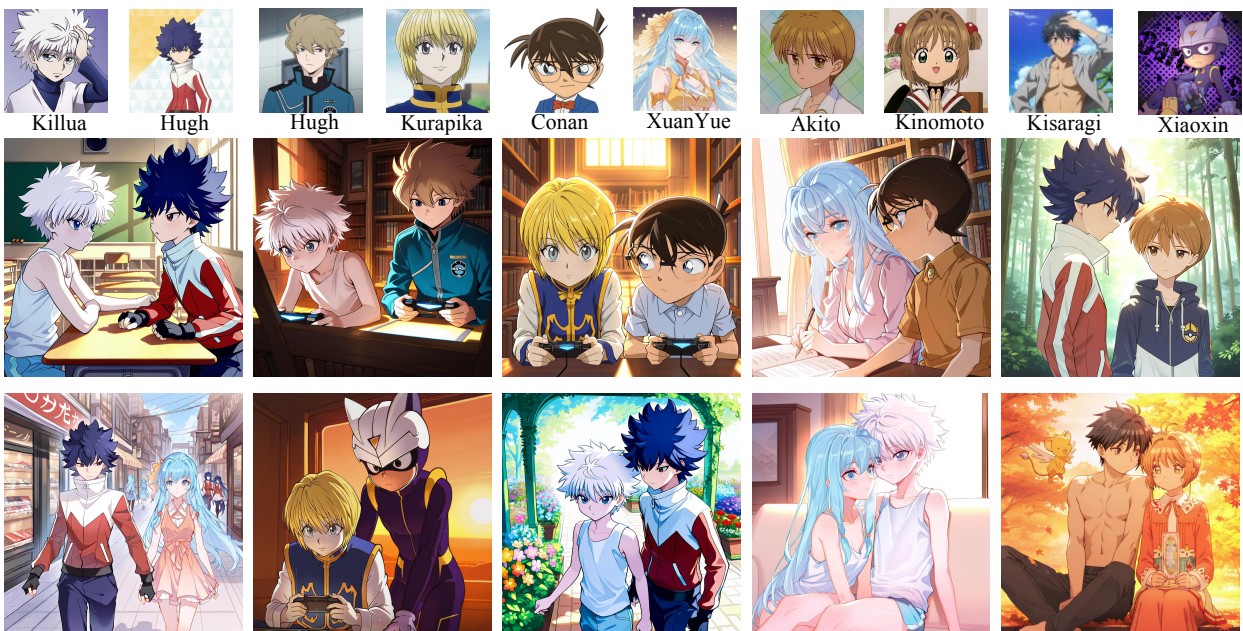

Figure 12: The results obtained from the SDXL-derived Illustrious-XL model further demonstrate the generalizability of our method.

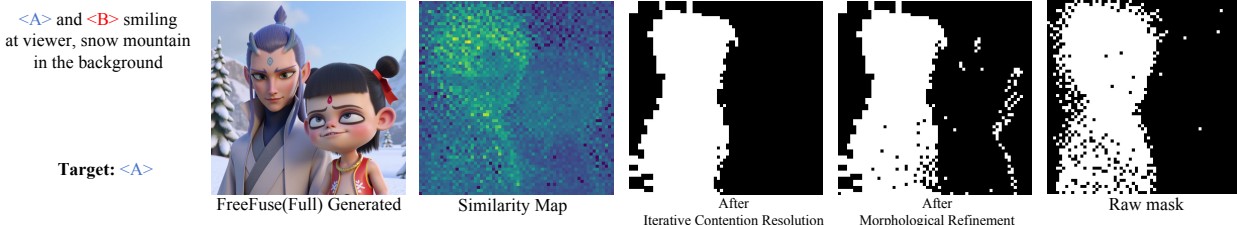

Figure 13: Post-processing plays a crucial role in further eliminating hollow artifacts within the masks and enhancing the overall stability of the method.

# B Post-processing Details

To translate the similarity maps into coherent and exclusive binary masks, we design a two-stage post-processing pipeline. This pipeline ensures topological integrity for the foreground regions and enforces strict semantic segregation between competing subjects.

## B.1 Morphological Refinement

To distinguish the foreground from the background, we apply a sequence of morphological operations to the binary foreground mask $M_{raw}$.

Let $\mathcal{K}$ denote the $2 \times 2$ structural element

$$\mathcal{K} = \begin{bmatrix} 1 & 1 \\ 1 & 1 \end{bmatrix}. \tag{8}$$

We define the *Opening* operation ($\circ$) to remove isolated noise pixels, followed by the *Closing* operation ($\bullet$) to fill interior holes:

Table 4: Hyperparameter settings for FreeFuse mask extraction and post-processing.

| Parameter | Symbol | Value |
|---|---|---|
| Mask Extraction Steps | $K$ | 5 |
| Inference Denoising Steps | $T_{\text{denoise}}$ | 28 |
| FreeFuseAttn Block | $b$ | 18 |
| FreeFuseAttn Temperature | $\tau$ | 4000 |
| TopK Anchor Ratio | $\rho$ | 10% |
| TopK Anchors | $k_{\text{anchor}}$ | $\max(1, \lfloor \rho N \rfloor)$ |
| Morphological Kernel | $\mathcal{K}$ | $\begin{smallmatrix} 1 & 1 \\ 1 & 1 \end{smallmatrix}$ |
| Router Iterations | $T_{\text{route}}$ | 15 |
| Momentum | $\mu$ | 0.2 |
| Gravity Weight | $\lambda_g$ | $2 \times 10^{-5}$ |
| Spatial Voting Weight | $\lambda_s$ | $2 \times 10^{-5}$ |

$$M_{opened} = M_{raw} \circ \mathcal{K} = (M_{raw} \ominus \mathcal{K}) \oplus \mathcal{K} \tag{9}$$

$$M_{clean} = M_{opened} \bullet \mathcal{K} = (M_{opened} \oplus \mathcal{K}) \ominus \mathcal{K} \tag{10}$$

where $\oplus$ and $\ominus$ denote Dilation and Erosion, respectively. This process yields a topologically clean foreground mask $M_{clean}$ that delineates the union of all subjects against the background.

## B.2 Iterative Contention Resolution (Router)

Within the foreground region, multiple subject LoRAs may compete for the same spatial tokens. To resolve these conflicts and assign each token $p$ to a unique subject $c \in \{1, \ldots, C\}$, we propose an Iterative Routing algorithm. Unlike standard argmax or Softmax, which suffer from "winner-takes-all" instability and lack spatial awareness, our method incorporates spatial cohesion constraints and adaptive load balancing.

We initialize the routing logits $L_{p,c}^{(0)}$ with the similarity scores derived from FreeFuseAttn. The algorithm updates these logits iteratively over $T$ steps (default $T = 15$). At each step $t$, the update rule is composed of four terms:

**1. Linear Normalization.** To prevent exponential suppression of weaker signals (a common issue with Softmax), we normalize logits linearly to the range $[0, 1]$:

$$P_{p,c}^{(t)} = \frac{L_{p,c}^{(t)} - \min_c(L_{p,c}^{(t)})}{\max_c(L_{p,c}^{(t)}) - \min_c(L_{p,c}^{(t)}) + \epsilon} \tag{11}$$

**2. Momentum Averaging.** We maintain a running average of these probabilities $\bar{P}^{(t)}$ with momentum $\mu = 0.2$ to stabilize the trajectory.

**3. Spatial Cohesion (Gravity).** We calculate the dynamic centroid $(\bar{x}_c, \bar{y}_c)$ for each subject based on the soft distribution $\bar{P}^{(t)}$. A penalty is applied proportional to the squared Euclidean distance from the centroid to enforce compactness:

$$\mathcal{E}_{gravity}(p, c) = \lambda_g \cdot \|\mathbf{u}_p - (\bar{x}_c, \bar{y}_c)\|^2 \tag{12}$$

where $\mathbf{u}_p$ is the spatial coordinate of token $p$ ranging from $-1$ to $1$.

**4. Local Neighborhood Voting.** To encourage local smoothness, we aggregate votes from the $3 \times 3$ neighborhood $\mathcal{N}(p)$:

$$\mathcal{V}_{spatial}(p, c) = \lambda_s \sum_{q \in \mathcal{N}(p)} \bar{P}_{q,c}^{(t)} \tag{13}$$

**Final Update Rule.** Combining these components, the logits are updated as:

$$L_{p,c}^{(t+1)} = L_{p,c}^{(t)} + \mathcal{V}_{spatial}(p,c) - \mathcal{E}_{gravity}(p,c) \tag{14}$$

After $T$ iterations, the final subject mask is obtained via $\arg\max_c L_{p,c}^{(T)}$. This ensures that the generated masks are not only semantically grounded but also spatially compact.

For reproducibility, we list the specific hyperparameters used in our experiments in Tab. 4.

Fig. 13 provides an intuitive example, demonstrating that our proposed post-processing plays a crucial role in enhancing the cohesion and stability of the masks.

## C  Prompts and LoRAs for Experiments

To ensure the reproducibility of our experiments, we provide the full list of scene prompts used in the multi-subject interaction stress tests. These prompts serve as the environmental context into which the specific subject tokens (e.g., `<Subject A>`, `<Subject B>`, etc.) are integrated.

---

**Scene Prompts for 4-Subject Generation**

1. Four characters talking on a riverside promenade lined with cherry trees, petals falling into the water, soft morning sunlight illuminating their smiles.

2. Four friends sitting on a rustic porch of a countryside cottage, chatting animatedly, dappled sunlight filtering through the hanging vines.

3. Four characters standing on a vast frozen lake in the wilderness, bundled in heavy coats, steam rising from their breath, golden hour lighting reflecting on the ice.

4. Four people having a serious conversation on a tree-lined city boulevard, surrounded by fallen orange and red leaves, overcast sky creating a moody atmosphere.

5. Four characters talking on a busy downtown street corner in the rain, huddled under umbrellas, wet asphalt reflecting neon signs, cinematic cool tones.

6. Four teenagers hanging out on a rooftop terrace overlooking the city, vibrant sunset colors painting the sky, nostalgic vibe.

7. Four characters walking down an ancient cobblestone alley, engaged in deep discussion, long shadows stretching out behind them in the late afternoon sun.

8. Four figures whispering on a mist-covered harbor dock at dawn, mysterious atmosphere, soft focus background of docked ships.

9. Four professionals in suits discussing business on a high-rise office balcony, modern city skyline in the background, bright clear blue sky.

10. Four characters having a picnic in a sunny wildflower meadow near a cliff edge, talking and laughing, surrounded by colorful blooming nature.

11. Four people arguing heatedly on a desolate windswept beach, hair blowing in the wind, storm clouds gathering above the ocean, dramatic high-contrast lighting.

12. Four characters talking by a stone fountain in an old European town square, vintage atmosphere, warm sunlight glowing on the water.

13. Four joggers taking a break on a scenic coastal road, dewdrops on the roadside grass, fresh and energetic atmosphere.

14. Four characters sitting in a circle on the sand of a quiet beach at night, illuminated by the glow of a nearby campfire, cozy and intimate setting.

15. Four silhouettes talking on a pedestrian bridge over a city canal at twilight, reflection visible in the water below, purple and blue gradient sky.

16. Four students studying and talking in a university library courtyard, seated under a large oak tree, scattered books, warm afternoon light.

17. Four characters talking while walking their dogs along a quiet suburban sidewalk, dynamic poses, bright and cheerful daylight.

18. Four elderly characters sitting at a table outside a street corner cafe, talking reminiscences, peaceful atmosphere, soft golden lighting.

19. Four characters chatting in a neon-lit back alley of a cyberpunk city, glowing holographic ads around them, rain-slicked surfaces.

20. Four curious characters looking at a map in a dense jungle clearing, sunbeams breaking through the thick canopy, adventure theme.

---

**Character-Object Interaction Prompts**

*Note: In these prompts, `<Subject>` is replaced by the specific character token.*

1. **Tactical Pistol:** `<Subject>` aiming A modern semi-automatic pistol with a threaded suppressor attachment, featuring a two-tone black and silver finish.

2. **Golden Snitch:** `<Subject>` reaching out hand to catch A gleaming golden snitch with intricately ribbed wings spread wide.

3. **Inception Top:** `<Subject>` staring intently at A sleek black spinning inception top balanced on its needle-sharp point spinning flawlessly on the table.

4. **Nike SB Sneaker:** `<Subject>` sitting on the ground and holding a Nike SB sneaker in classic black, white, and red colorway.

5. **Armored Battle Car:** `<Subject>` standing casually in front of A heavily armored tactical vehicle with angular matte gray plating.

---

To facilitate reproducibility and ensure transparency, we list the 15 LoRA checkpoints utilized in our quantitative and qualitative comparisons. The dataset consists of 10 Character LoRAs and 5 Object LoRAs, selected to cover a diverse range of styles (photorealistic, anime, 3D) and structural complexities (rigid bodies, deformable objects), as illustrated in Fig. 14 and Fig. 15.

All checkpoints were obtained from open-source repositories (e.g., Civitai, Hugging Face). Upon the publication of this paper, we will release the specific weight files and the corresponding retrieval scripts to the community.

| | | | | | |
|---|---|---|---|---|---|
| | Alcina Dimitrescu, A striking, pale-skinned aristocratic woman wearing a wide-brimmed black hat, a pearl necklace featuring a bee pendant, and a low-cut ivory dress, with bold red lipstick and dark | Real | | Greggs, A handsome young man with thick, wavy brown hair, striking brown eyes, subtle stubble, and noticeable dimples, wearing a formal dark suit and a black bow tie | Real |
| | Neytiri, a Na'vi woman with bright yellow eyes, distinct blue skin with bioluminescent markings, pointed ears with gauges, and a simple beaded choker | Humanoid | | RanMōri, anime_style, a young animated woman with long dark hair, featuring a distinctive point at the top, wearing a pink dress with a white floral pattern and a white-collared shirt underneath | Anime |
| | Leela, a South Asian woman with long, dark, wavy hair, adorned in traditional gold jewelry, including intricate earrings and a layered necklace. She wears a small black bindi and a nose piercing, with soft, elegant makeup | Real | | Anime Detective Conan, a young boy with slightly messy dark brown hair with a small cowlick, large blue eyes, wearing round glasses, wears a blue school blazer with a white shirt and red bow tie, gray shorts, white socks, and red sneakers | Anime |
| | Aobing, a sleek, androgynous CGI animated male character with pale lavender hair pulled into a high topknot, delicate blue-gray horns, a matching symbol on their forehead, long, pointed ears, wearing an elegant, light-colored robe | 3D | | Nezha, a highly-detailed 3D Pixar-style cartoon character of a young boy with black hair styled in twin buns tied with bright red ribbons, wears a red outfit with a chunky gold necklace | 3D |
| | AliceWaifu, a young, 3D CGI girl resembling Alice, featuring bright blonde hair, large, expressive blue eyes, a cute, stylized face with rosy cheeks, and wearing a classic blue dress with a white pinafore and a dark blue headband bow | 3D | | Naruto uzumaki, with signature spiky blonde hair and blue eyes, wearing a sleek black and red techwear hoodie and modern red over-ear headphones integrated with his Hidden Leaf headband. | Anime |

Figure 14: Character LoRAs used in quantitative and qualitative comparisons.

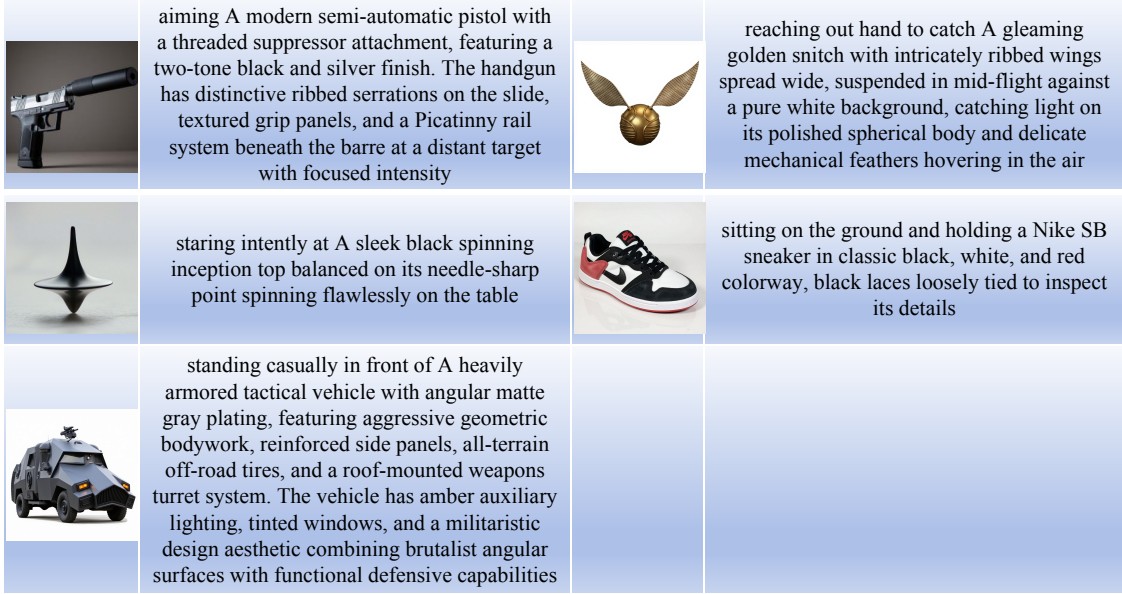

| | | | |
|---|---|---|---|
| | aiming A modern semi-automatic pistol with a threaded suppressor attachment, featuring a two-tone black and silver finish. The handgun has distinctive ribbed serrations on the slide, textured grip panels, and a Picatinny rail system beneath the barre at a distant target with focused intensity | | reaching out hand to catch A gleaming golden snitch with intricately ribbed wings spread wide, suspended in mid-flight against a pure white background, catching light on its polished spherical body and delicate mechanical feathers hovering in the air |
| | staring intently at A sleek black spinning inception top balanced on its needle-sharp point spinning flawlessly on the table | | sitting on the ground and holding a Nike SB sneaker in classic black, white, and red colorway, black laces loosely tied to inspect its details |
| | standing casually in front of A heavily armored tactical vehicle with angular matte gray plating, featuring aggressive geometric bodywork, reinforced side panels, all-terrain off-road tires, and a roof-mounted weapons turret system. The vehicle has amber auxiliary lighting, tinted windows, and a militaristic design aesthetic combining brutalist angular surfaces with functional defensive capabilities | | |

Figure 15: Object LoRAs used in quantitative and qualitative comparisons.

## D   More Qualitative Comparisons

We further provide more qualitative comparisons in Fig. 16.

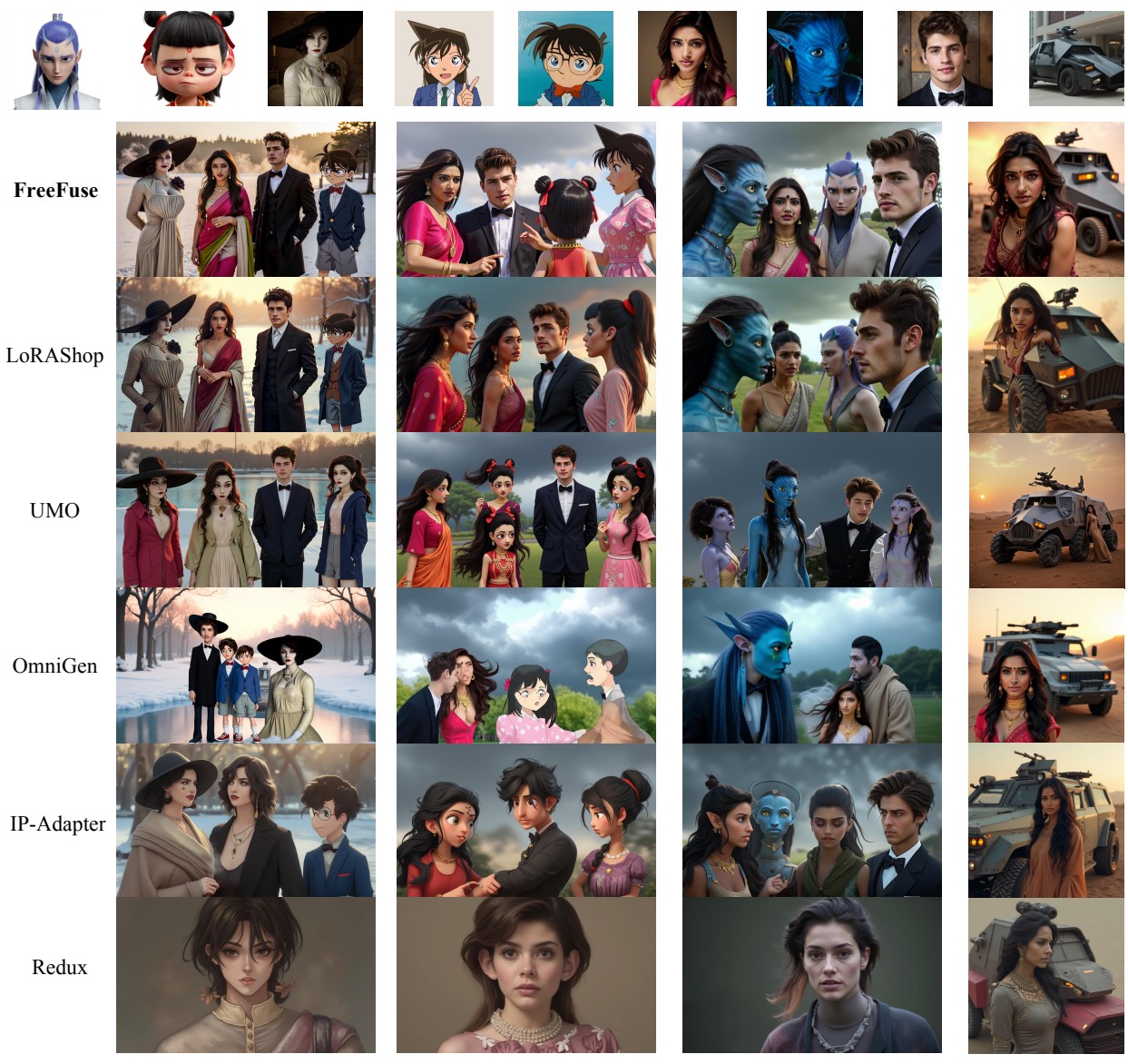

Figure 16: Additional Qualitative Comparisons.

# E    Scaling to More LoRAs

To evaluate whether FreeFuse remains effective as the number of simultaneously loaded LoRAs increases, we construct a controlled LoRA-count ablation with $k \in \{2, 3, 4, 5\}$ subject LoRAs. The baseline uses the same LoRAs and prompts with naive simultaneous LoRA activation, while FreeFuse uses the proposed mask extraction, routing, and attention bias. As shown in Tab. 5, identity preservation becomes increasingly difficult as more adapters compete, but FreeFuse consistently improves ArcFace ID over the baseline at every LoRA count, with an average gain of +0.3207. HPSv2 remains comparable or slightly higher, indicating that the identity gain does not come from degrading global image preference.

Table 5: Quality trends as the number of simultaneously loaded LoRAs increases. Baseline denotes direct multi-LoRA activation without FreeFuse routing.

| # LoRAs | ArcFace ID ↑ | | | HPSv2 ↑ | | |
|---|---|---|---|---|---|---|
| | Baseline | FreeFuse | Δ | Baseline | FreeFuse | Δ |
| 2 | 0.4305 | **0.7002** | +0.2697 | 0.2806 | **0.2806** | +0.0000 |
| 3 | 0.2695 | **0.6623** | +0.3928 | 0.2786 | **0.2804** | +0.0019 |
| 4 | 0.2028 | **0.6285** | +0.4257 | 0.2776 | **0.2809** | +0.0033 |
| 5 | 0.1759 | **0.3705** | +0.1946 | 0.2712 | **0.2773** | +0.0061 |
| Avg. | 0.2697 | **0.5904** | +0.3207 | 0.2770 | **0.2798** | +0.0028 |

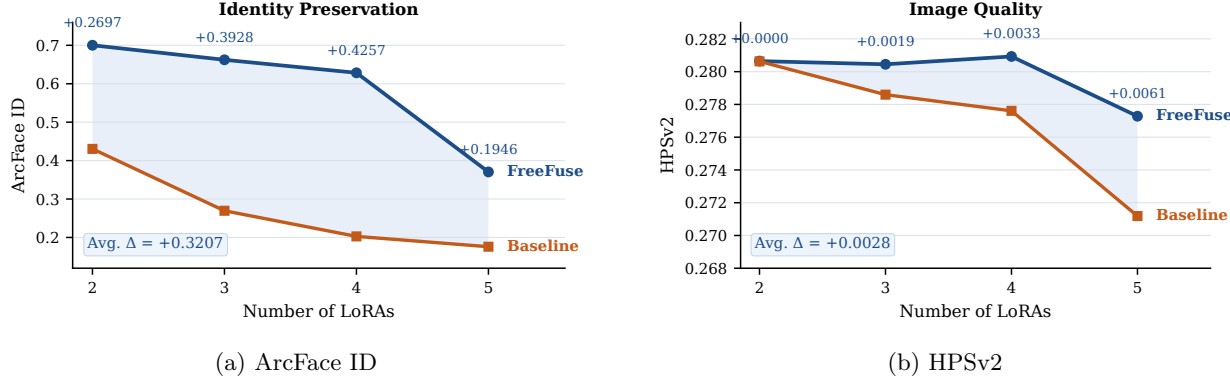

(a) ArcFace ID

(b) HPSv2

Figure 17: Trends under increasing LoRA count. FreeFuse preserves identity substantially better than direct multi-LoRA activation, while maintaining comparable global preference scores.

# F    Additional Evaluation on a Public Dual-Subject Protocol

We further evaluate FreeFuse on the public dual-subject protocol of LoRAShop (Dalva et al., 2025), whose LoRA weights and prompting setup are directly reproducible. Tab. 6 reports the comparison under this protocol. FreeFuse achieves the best identity preservation and prompt alignment, while remaining competitive in human-preference and aesthetic metrics. This protocol is complementary to our main benchmark: LoRAShop primarily evaluates two real-person LoRAs, whereas our main benchmark includes photorealistic, anime, 3D, object, garment, and accessory LoRAs, and stresses up to four simultaneous LoRAs in a single prompt.

Table 6: Evaluation on the public LoRAShop dual-subject protocol.

| Method | ArcFace ID ↑ | CLIP-T bigG ↑ | HPSv2 ↑ | Aesthetic v2 ↑ |
|---|---|---|---|---|
| **FreeFuse** | **0.6557** | **0.5184** | 0.3117 | 5.7426 |
| IP-Adapter | 0.2139 | 0.4826 | 0.3100 | **6.2833** |
| OmniGen | 0.5554 | 0.5032 | 0.3040 | 6.0466 |
| LoRAShop | 0.6376 | 0.5157 | **0.3153** | 5.9921 |
| FLUX Redux | 0.2478 | 0.2508 | 0.2527 | 6.1375 |
| UMO | 0.5091 | 0.4991 | 0.3070 | 6.0185 |

## G  User Study Details

To complement our quantitative metrics, we conducted a blind user preference study to evaluate the perceptual quality of the generated images. The study focused on two core capabilities: multi-subject interaction (Q1) and character-object interaction (Q2). It includes 17 participants, aged 18-40, spanning undergraduate students, master's students, PhD students, and professors. The questionnaire uses blind presentation for all methods, and the study contains 40 cases in total (20 character-character and 20 character-object cases). The final score is obtained by averaging participant rankings across cases.

Participants were presented with anonymized outputs from FreeFuse and baseline methods (shuffled to prevent bias) and were asked to perform a ranking task. The questionnaire instructions and evaluation criteria were designed as follows:

---

**Questionnaire Instruction**

**Task:** Please rank the set of generated images displayed below. The reference character images are provided as `<A>`, ``, `<C>`, and `<D>`.

**Ranking Criteria (in descending order of priority):**

1. **Identity Preservation:** How accurately does the generated character resemble the target reference image? (Higher consistency = Higher Rank)

2. **Text-Image Alignment:** Does the image accurately reflect the content described in the prompt? (Higher consistency = Higher Rank)

3. **Aesthetic Quality:** How visually appealing and natural is the overall image? (Higher quality = Higher Rank)

**Example Prompt (Q1 - Multi-Character):** *"<A>, , <C>, and <D> are having a friendly conversation by the fountain pool."*

---

For the Character-Object interaction (Q2), the same criteria were applied, with "Identity Preservation" extending to the structural accuracy of the specific object LoRA. The aggregated rankings were converted into the preference scores reported in Tab. 1.

## H  Latency and Computational Cost Analysis

To evaluate the practical efficiency of FreeFuse, we measured the inference latency across varying numbers of subject LoRAs (from 2 to 6). All experiments were conducted on a single L20 (48GB). The baseline represents the standard FLUX.1-dev inference with naive LoRA sampling.

**Experimental Protocol.** We define the total inference time as the duration from the initial prompt encoding to the final pixel decoding. For FreeFuse, this includes the additional Phase 1 (Mask Extraction) and the Iterative Contention Resolution.

Table 7: Inference latency comparison (in seconds) at $1024 \times 1024$ resolution. **Comparison Key:** While Naive Flux is faster, it fails to generate coherent images. Compared to the leading training-free competitor LoRAShop Dalva et al. (2025), FreeFuse achieves a $1.8\times \sim 2.1\times$ speedup, offering a superior trade-off between fidelity and efficiency.

| Method | 2 LoRAs | 3 LoRAs | 4 LoRAs | 5 LoRAs | 6 LoRAs |
|---|---|---|---|---|---|
| Naive Flux (Fails on complex scenes) | 21.95s | 24.60s | 27.11s | 29.64s | 32.08s |
| LoRAShop Dalva et al. (2025) | 64.78s | 84.36s | 103.60s | 122.96s | 142.27s |
| **FreeFuse (Ours)** | **36.08s** | **42.96s** | **50.29s** | **58.63s** | **67.96s** |
| **Speedup vs. LoRAShop** | **+44.3%** | **+49.1%** | **+51.5%** | **+52.3%** | **+52.2%** |

**Note on Efficiency Trade-offs:** While the naive Flux baseline exhibits lower latency, it fails to produce semantically valid outputs in multi-subject scenarios due to severe feature conflict (as evidenced in Fig. 1). Consequently, the computational overhead of FreeFuse is a necessary trade-off for ensuring identity preservation. Furthermore, when benchmarked against LoRAShop Dalva et al. (2025), the leading training-free alternative, FreeFuse demonstrates significantly better scalability. As the number of subjects increases to six, LoRAShop's inference time rises linearly to over 140 seconds, whereas FreeFuse remains under 70 seconds. This indicates that our framework provides a more practical balance between high-fidelity generation and runtime efficiency.

