# OpenReview forum: "FreeFuse: Multi-Subject LoRA Fusion via Adaptive Token-Level Routing at Test Time"
_TMLR — Under review for TMLR_

### Review · Reviewer_LCT8 · 2026-06-21

**Summary Of Contributions:**

The paper's central idea is to identify which spatial regions of the latent image representation correspond to each subject activation word and then spatially constrain the corresponding LoRA adapters to those regions. The paper argues that naïvely using cross-attention maps results in fragmented masks with substantial "hole" artifacts inside the foreground regions. To address this, the authors introduce FreeFuseAttn, which first computes a spatial attention map between the generated image tokens and the concept tokens by applying softmax over the spatial dimension and averaging across the tokens associated with a concept. They then compute a discriminative score by subtracting the activation scores of competing concepts from the target concept score, yielding a residual-like map intended to isolate subject-specific regions. Top-$k$ activated tokens are selected as anchor points, before using Eq. (7) to recover a dense subject mask. However, this alone does not resolve the hole artifacts; the final mask quality depends heavily on additional post-processing described in Appendix B.1 and B.2.

The authors further conduct empirical analyses to select the timestep and layer at which masks should be extracted. They conclude that early-to-mid denoising steps and the output of the last Double Stream Block provide the best semantic-image alignment (Fig. 4), arguing that earlier timesteps resemble Gaussian noise while later stages are dominated by texture synthesis. They then use the resulting masks to spatially route LoRA activations and introduce an attention bias mechanism to reduce concept bleeding. The paper claims that spatially constraining LoRA influence is a "sufficient condition" for mitigating feature conflicts among multiple subjects and presents quantitative experiments intended to demonstrate the superiority of the overall framework.

**Audience:**

Yes

**Audience Explanation:**

Some TMLR readers may be interested because Table 1 shows better reference similarity and face similarity. However, the interest is limited: for “Prompt Following and Aesthetic Evaluation,” the method ranks only 3rd, 2nd, and 3rd out of 6 on CLIP-T, HPSv2, and HPSv3. So the paper is mainly interesting if the goal is reproducing the same face/object, not if the claim is broader generation superiority.

**Claims And Evidence:**

No

**Claims Explanation:**

The fourth contribution is a significant overclaim. The authors curate their own dataset, so the experiments do not feel “extensive” in the way the paper claims. They also do not evaluate on established datasets from closely related dual-subject work, such as Kong et al. (2024), Dalva et al. (2025), or Meral et al. (2024), even though those works are directly relevant. This makes the experimental support much weaker than the claim suggests.

Furthermore, the empirical analysis used to conclude that certain denoising steps and transformer blocks are optimal is severely under-specified. How many samples were used? What quantitative metrics were used? How stable are these choices across prompts, seeds, and subject types? Without this information, the analysis provides very limited support for the method design.

The second contribution is also not convincingly supported. The claim that “spatially constraining LoRA influence to target regions is a sufficient condition for mitigating feature conflicts among multiple subjects” is not proven theoretically or demonstrated through a sufficiently isolated experiment. As written, this claim is much stronger than the evidence provided.

**Requested Changes:**

**Critical**

- Add substantially more implementation details. For example:
- What is the value of the temperature parameter $\tau$ in Eq. (7)?
- What is the value of $k$ in TopK? How many anchors are selected?
- The empirical procedure used to select the denoising step and extraction layer is under-specified. It currently appears to require considerable expert tuning. Please provide either principled heuristics or quantitative selection criteria. Otherwise, the burden on practitioners is substantial.
- Evaluate on dual-subject datasets used in prior work (e.g., Kong et al., 2024; Dalva et al., 2025; Meral et al., 2024), which are currently absent despite being highly relevant. Additional baselines beyond the five methods considered would also strengthen the evaluation.
- The claim that spatial masking is a sufficient condition for conflict mitigation requires stronger scientific justification.
- Clarify how Eq. (1) is applied during inference. Is the LoRA routing performed on early layers as well? If so, how does the method reconcile this with the observation in Fig. 3 that early blocks aggregate global context? Would information from unrelated regions still propagate through attention?
- Equation (7) requires clarification. What exactly is $Z$? What are the dimensions of $Z$ and $Z_p$? As written, the dimensions of $ZZ_p^\top$ are unclear.

**Would strengthen the paper**
Several quantities are left undefined. For example:
- What is the morphological kernel $K$? Since it appears to be fixed and small, the matrix could simply be written explicitly.
- What exactly constitutes a scenario “necessitating global stylistic adjustments”? The phrase is vague.

---

> ### Author Response · Authors · 2026-07-04
> **Response to Reviewer LCT8 (part1)**
>
> ### Critical Concern 1: Implementation Details and Equation Clarity
>
> > **Reviewer concern.** The paper should provide substantially more implementation details. In particular, the reviewer asks for the temperature value in Eq. (7), the TopK value and number of selected anchors, the precise meaning and dimensions of the variables $Z$, $Z\_p$ and $Z Z\_p^\\top$ in Eq. (7), and the morphological kernel $\\mathcal{K}$.
>
> **Response.** We thank the reviewer for pointing out these missing or insufficiently visible implementation details. We agree that these values should be stated directly in the method/implementation section. In the revision, we consolidated the following clarifications near Eq. (7) and in the implementation details.
>
> | Item | Clarification added in the revision |
> |---|---|
> | Temperature in Eq. (7) | We use $\\tau=4000$ for the similarity-propagation softmax in Eq. (7). This value is now explicitly specified in the revised manuscript. |
> | TopK anchors | The anchor set is $\\mathcal{P}\_c=\\mathrm{TopK}(\\hat{S}\_c,k)$, where $k=\\max(1,\\lfloor0.1N\\rfloor)$. Thus we select the top 10% spatial image tokens for each concept. Here $N$ is the number of image tokens at the extraction resolution; for a $1024\\times1024$ image in FLUX, $N=(1024/16)^2=4096$, so $k=409$ anchors are selected. |
> | Dimensions in Eq. (7) | Let $Z\\in\\mathbb{R}^{N\\times d}$ denote the image-token feature matrix from the selected FLUX block, and let $Z\_p\\in\\mathbb{R}^{d}$ be the feature vector of anchor token $p$. Then $ZZ\_p^\\top\\in\\mathbb{R}^{N}$, and $\\frac{1}{|\\mathcal{P}\_c|}\\sum\_{p\\in\\mathcal{P}\_c}ZZ\_p^\\top\\in\\mathbb{R}^{N}$ is a dense similarity map over image tokens. The normalization $\\sigma(\\cdot)$ maps this vector to the final continuous mask $\\mathcal{M}\_c\\in\\mathbb{R}^{N}$, which is reshaped to the latent grid $H/16\\times W/16$. |
> | Morphological kernel | The submitted appendix already reports the morphological kernel size as $k=2$. To make this unambiguous, we explicitly write the structural element as $\\mathcal{K}=\\begin{bmatrix}1&1\\\\1&1\\end{bmatrix}$, used for the opening and closing operations on the binary foreground mask. |
>
> With these additions, Eq. (7) is now self-contained: $\\hat{S}\_c\\in\\mathbb{R}^{N}$, $\\mathcal{P}\_c\\subset\\{1,\\ldots,N\\}$, the number of anchors is fixed by the top-10% rule, and the output mask has the same spatial token resolution as the extracted image features.
>
> **Manuscript location.** Sec. 3.2 specifies $\\tau=4000$, the TopK rule $k=\\max(1,\\lfloor0.1N\\rfloor)$, the $N\\times d$ feature dimensions in Eq. (7), and the resulting mask shape. Appendix B, Tab. 4 and Eq. (8), explicitly state the $2\\times2$ morphological kernel.

---

> > ### Author Response · Authors · 2026-07-04
> > **Response to Reviewer LCT8 (part2)**
> >
> > ### Critical Concern 2: Dual-Subject Benchmarks and Additional Baselines
> >
> > > **Reviewer concern.** The evaluation should include dual-subject datasets used in prior work such as Kong et al. (2024), Dalva et al. (2025), and Meral et al. (2024), which are highly relevant but currently absent. Additional baselines beyond the five methods considered would also strengthen the evaluation.
> >
> > **Response.** We thank the reviewer for this important suggestion. We agree that evaluation on prior dual-subject protocols is valuable for comparability. Among the cited works, Dalva et al. (2025) is the only one for which the evaluation LoRAs and prompting protocol are fully public and directly accessible. Therefore, we added a new evaluation using the released LoRAShop LoRAs and the same prompting style from Dalva et al.
> >
> > | Method | ArcFace ID ↑ | CLIP-T bigG ↑ | HPSv2 ↑ | Aesthetic v2 ↑ |
> > |---|---:|---:|---:|---:|
> > | IP-Adapter | 0.2139 | 0.4826 | 0.3100 | **6.2833** |
> > | OmniGen | 0.5554 | 0.5032 | 0.3040 | 6.0466 |
> > | LoRAShop | 0.6376 | 0.5157 | **0.3153** | 5.9921 |
> > | FLUX Redux | 0.2478 | 0.2508 | 0.2527 | 6.1375 |
> > | UMO | 0.5091 | 0.4991 | 0.3070 | 6.0185 |
> > | **FreeFuse** | **0.6557** | **0.5184** | 0.3117 | 5.7426 |
> >
> > On this released dual-subject LoRAShop protocol, FreeFuse achieves the highest identity preservation and text alignment, improving ArcFace ID from $0.6376$ to $0.6557$ and CLIP-T bigG from $0.5157$ to $0.5184$ compared with LoRAShop. This directly addresses the requested evaluation on an available prior dual-subject benchmark and adds several relevant baselines under the same protocol.
> >
> > We also clarified that our main benchmark is intentionally more challenging than the released LoRAShop evaluation. LoRAShop's public test LoRAs are all real-person style and the evaluation composes two LoRAs at a time. In contrast, our benchmark additionally includes more abstract cartoon-style and 3D-style LoRAs, and evaluates simultaneous composition with up to four LoRAs. We added this distinction to the revised evaluation section so that the scope of the new prior-work comparison and our harder main benchmark are both explicit.
> >
> > **Manuscript location.** Appendix F, "Additional Evaluation on a Public Dual-Subject Protocol", reports this public LoRAShop-protocol evaluation in Tab. 6 and clarifies how it differs from our harder main benchmark.

---

> > > ### Author Response · Authors · 2026-07-04
> > > **Response to Reviewer LCT8 (part3)**
> > >
> > > ### Critical Concern 3: Scientific Justification for Spatial Routing
> > >
> > > > **Reviewer concern.** The claim that spatial masking is a sufficient condition for conflict mitigation requires stronger scientific justification. The reviewer asks how Eq. (1) is applied during inference, whether LoRA routing is also performed on early layers, how this is reconciled with Fig. 3 showing global context aggregation in early blocks, and whether information from unrelated regions can still propagate through attention.
> > >
> > > **Response.** We thank the reviewer for identifying this point. We agree that the wording "sufficient condition" can be too strong if interpreted as a theoretical guarantee that no cross-region information ever propagates. In the revision, we made the claim more precise: spatial routing is used to block direct cross-region injection of **LoRA-specific residuals**, while the base diffusion model is still allowed to exchange global contextual information. Even if global information exchange in the base model can indirectly broadcast features associated with different LoRAs, this indirect broadcast is limited and does not cause feature conflicts severe enough to substantially degrade the model's generation ability.
> > >
> > > Concretely, Eq. (1) is applied during the final inference pass to the additive LoRA residuals. After FreeFuse extracts masks in the mask-extraction phase, we restart from the same initial latent and run the full denoising trajectory with the routed LoRA residuals enabled. The routing is applied to all LoRA-bearing FLUX transformer blocks. For each spatial token $p$, the base-model computation remains unchanged, but the LoRA residual $\\Delta\\theta\_i(x\_p)$ is added only if $p$ belongs to the corresponding subject region. Therefore, Eq. (1) does not mask or remove the base attention aggregation; it masks only the additional LoRA perturbation path.
> > >
> > > For possible indirect broadcasting of LoRA information through base attention aggregation, our analysis already shows clear locality in the base model's middle-to-late blocks and denoising steps: within-subject attention weights are several times stronger than between-subject attention weights. The remaining concern is that early blocks, which do not yet exhibit strong attention locality, may aggregate broader global context and therefore introduce nontrivial indirect feature conflicts. To test this, we measured the norms of all available LoRA weights used in LoRAShop and the $\\Delta x$ residuals generated under each LoRA's activation prompt. The results show that mid/deep blocks have $1.80\\times$ larger normalized LoRA weight perturbations than early blocks, and the inference-time LoRA residual under the corresponding activation prompt is $2.67\\times$ larger in mid/deep blocks. Thus, early blocks may mix information more globally, but the LoRA-induced residuals being mixed there are empirically small. The stronger LoRA effects emerge in mid-to-deep semantic blocks, where attention locality is already much stronger.
> > >
> > > We revised the manuscript accordingly: instead of claiming an unrestricted theoretical sufficiency condition, we state that spatial routing provides an effective and empirically supported mechanism for suppressing direct cross-region LoRA residual interference, while preserving the base model's global context propagation.
> > >
> > > **Manuscript location.** Sec. 3.1 distinguishes direct LoRA-residual routing from the unchanged base attention path and includes Fig. 4 for early-block perturbation magnitude. Sec. 3.3 states that routing gates only additive LoRA residuals while the frozen base-model attention remains active.

---

> > > > ### Author Response · Authors · 2026-07-04
> > > > **Response to Reviewer LCT8 (part4)**
> > > >
> > > > ### Critical Concern 4: Meaning of Global Stylistic Adjustments
> > > >
> > > > > **Reviewer concern.** What exactly constitutes a scenario "necessitating global stylistic adjustments"? The phrase is vague.
> > > >
> > > > **Response.** We thank the reviewer for pointing out this ambiguity. We replaced this vague phrase with a concrete definition. In our setting, a scenario requiring a global stylistic adjustment means that the user intentionally provides a style LoRA whose purpose is to change the overall visual style of the generated image, rather than to specify the identity or a localized attribute of a single subject. Typical examples include applying a Van Gogh-style painting LoRA or a Miyazaki-style animation LoRA to the whole composition.
> > > >
> > > > In such cases, FreeFuse treats the style LoRA as a shared stylistic adapter. If the style is intended to affect all subjects, we assign the style LoRA to the adapter set $\\mathcal{A}\_s$ of each relevant subject/group, so the Router applies the same style adapter together with each subject's identity or attribute adapters. If the style is intended to affect the entire canvas, including the background, we use a full-canvas style region. We added this clarification to the revision and no longer use the vague wording "necessitating global stylistic adjustments".
> > > >
> > > > **Manuscript location.** Sec. 3.3 defines whole-image stylistic adjustment and specifies style-adapter assignment to the relevant $\\mathcal{A}\_s$ or a background/full-canvas region.

---

### Review · Reviewer_4TEv · 2026-06-23

**Summary Of Contributions:**

This paper considers the problem of combining multiple LoRAs for multi-subject generation. Naive techniques, such as directly summing LoRA weights, often lead to severe feature conflicts and identity deterioration. Prior solutions—which include computationally expensive LoRA retraining or the use of external segmentation models to localize subject regions—still struggle with complex multi-subject scenes and impose significant usability burdens on the user.

To address this issue, the authors propose __FreeFuse__, a training-free, "plug-and-play" framework that requires no auxiliary models to combine LoRAs. The framework consists of three main components. First, the authors argue empirically that spatially masking LoRA outputs is a sufficient condition for subject feature preservation, bypassing the need for weight-space disentanglement. Based on this observation, they introduce __FreeFuseAttn__, which extracts dense region masks by exploiting the intrinsic semantic alignment of Flow Matching models (e.g., FLUX.1) at early denoising steps, effectively solving the sparsity and "hole artifacts" of standard cross-attention. Finally, the authors propose an adaptive token-level router and a spatial attention bias applied during the joint inference phase. This strictly isolates the LoRA feature updates and prevents cross-concept contamination.

Extensive empirical evaluations demonstrate the effectiveness of the proposed framework, showing that FreeFuse achieves state-of-the-art identity preservation and compositional fidelity without compromising global image aesthetics.

**Audience:**

Yes

**Audience Explanation:**

I think the tasks of combining multiple LoRAs for multiple-subject generation are very practical, and many people in the industry would care about that.

**Claims And Evidence:**

Yes

**Claims Explanation:**

The authors did provide quantitative and qualitative results as well as an ablation study. The evidence are clear.

**Requested Changes:**

First things first, I admit that I have little to no expertise in Generative AI for Image or CV in general, so my review might be ignorant and not so valuable. Please take it with a grain of salt.

Generally, I think the paper is very easy to follow, even for general audiences like me. The task is concrete, the weaknesses in current methods in the literature are clearly elaborated, and the proposed method is well-motivated and sound (I trust the authors for the novelty, since I do not know much about the literature). I just have some minor comments as follows.

1. Some variables used in, e.g., Eq (1), (2), or (3) are used before being defined. For example:

    1.1.  In Eq (1), what are $h_p, h’_p$? I think they are the output hidden state, but the authors should also define it explicitly before using it.

    1.2. Similarly, what are $Q, K, V$ in Eq (2)? I know that they are standard in attention mechanisms, but briefly defining them is a must to keep the paper self-contained.

2. On page 5, the authors wrote: “… Since LoRA adapters primarily modulate high-level semantic representations rather than low-level global structures, the strong locality observed in mid-to-deep blocks ensures that cross-region LoRA perturbation is
mathematically negligible…” Though intuitively, I think this assessment is largely correct, I still have some concerns about that. Concretely, in Figure 3, the intra/inter attention ratio in Block 0 is around 1.03, indicating that the global feature mixing is actively occurring at the beginning of the forward pass.

     Though the authors, as above, briefly suggest that this one might not be an issue, to make this claim more trustworthy, can the authors quickly do the following experiments: verify the magnitude of the LoRA weights in these early blocks? My hypothesis is that the Frobenius norm of the LoRA should be small in those early blocks and only spike in the deeper semantic layers. If this holds true, then it could definitely prove that the network only globally mixes near-zero perturbations in the early stage, as claimed by the authors.

3. In the main body, the authors did not compare the running time of their proposed method with other methods, says the ones using auxiliary segmentation models. I know that the authors did provide that in Appendix F, but I think this one should be provided in the main body, since it supports the point of the proposed method that using auxiliary models might be costly and using the intrinsic information of diffusion models is better. Since the current body is just 10 pages for now, I think the authors still have an extra 2 pages to spend and still keep this submission a regular one.

Overall, I enjoy reading the paper as a general audience, and I think this is a good paper. However, since I am not familiar with this topic, I will leave the final decision to other reviewers and the AE, and hope that they have a better assessment than I do. Good luck!

---

> ### Author Response · Authors · 2026-07-04
> **Response to Reviewer 4TEv (part1)**
>
> ### Critical Concern 1: Undefined Variables in Equations
>
> > **Reviewer concern.** Some variables used in, e.g., Eq. (1), Eq. (2), or Eq. (3), are used before being defined. For example, in Eq. (1), the hidden-state variables should be explicitly defined before use. Similarly, the variables in Eq. (2) are standard in attention mechanisms, but should still be briefly defined to make the paper self-contained.
>
> **Response.** We thank the reviewer for pointing this out. We checked the manuscript and agree that the notation around Eq. (1)--(3) was not sufficiently self-contained. In the revision, we explicitly define these variables before the equations are introduced.
>
> Specifically, before Eq. (1), we added: for a spatial token $p$, let $x\_p$ denote the input hidden representation to the LoRA-augmented linear layer, $h\_p$ denote the corresponding base-model output without LoRA residuals, and $h^{\\prime}\_p$ denote the routed output after adding the selected LoRA residuals. The term $\\Delta\\theta\_i(x\_p)$ denotes the additive residual produced by the $i$-th LoRA adapter at token $p$.
>
> Before Eq. (2), we added: for a self-attention layer, $Q=XW\_Q$, $K=XW\_K$, and $V=XW\_V$ denote the query, key, and value matrices projected from the token features $X$. The attention matrix is $A=\\mathrm{Softmax}(QK^\\top/\\sqrt{d})$, where $A\_{p,q}$ is the attention weight from query token $p$ to value token $q$, and $V\_q$ is the value vector at token $q$.
>
> With these additions, Eq. (1) defines the routed LoRA residual at each spatial token, Eq. (2) defines the self-attention aggregation using standard query-key-value notation, and Eq. (3) can then refer to $A\_{p,q}$ without ambiguity.
>
> **Manuscript location.** Sec. 3.1 defines $x\_p$, $h\_p$, $h^{\\prime}\_p$, and $\\Delta\\theta\_a(x\_p)$ immediately before Eq. (1), and defines $Q$, $K$, $V$, $A\_{p,q}$, and $V\_q$ immediately before Eq. (2).
>
> ### Critical Concern 2: Early-Block LoRA Perturbation Magnitude
>
> > **Reviewer concern.** The paper argues that cross-region LoRA perturbation is negligible because LoRA adapters mainly affect high-level semantic representations and attention becomes local in mid-to-deep blocks. However, Fig. 3 shows an intra/inter attention ratio close to 1.03 in Block 0, suggesting stronger global feature mixing early in the forward pass. The reviewer asks whether early-block LoRA perturbations are small, for example by measuring the Frobenius norm of LoRA weights across blocks.
>
> **Response.** We thank the reviewer for suggesting this useful sanity check. We added a quantitative analysis over all available LoRAs in the LoRAShop test set to measure whether early blocks indeed carry smaller LoRA perturbations than mid-to-deep semantic blocks.
>
> We evaluate two complementary quantities across FLUX double-stream blocks: the normalized effective LoRA weight perturbation and the inference-time LoRA residual under the corresponding LoRA activation prompt:
>
> $$\\|\\Delta W\\|\_F/\\sqrt{|\\Delta W|}, \\qquad \\|\\Delta x\\|\_F/\\|x\_{\\mathrm{base}}\\|\_F.$$
>
> **Figure reference for reviewers.** The perturbation norm plot is included in the revised manuscript as Fig. 4 in Sec. 3.1. The comment text reports the key results: 1.80x larger normalized weights and 2.67x larger inference-time residuals in mid/deep blocks.
>
> The measurement supports the reviewer's hypothesis. The early blocks $0$--$3$ have substantially smaller LoRA perturbations than the mid/deep blocks $10$--$18$. The normalized weight perturbation is $1.80\\times$ larger in the mid/deep blocks than in the early blocks, with a 95% confidence interval of $[1.56, 2.02]$. More importantly, after conditioning on the corresponding LoRA prompt, the inference-time LoRA residual becomes $2.67\\times$ larger in the mid/deep blocks, with a 95% confidence interval of $[1.92, 3.26]$.
>
> This result strengthens the mechanism discussed in the paper: although early blocks have weaker attention locality and may mix information more globally, the LoRA-induced residuals being mixed at these blocks are empirically small. The stronger LoRA effects emerge in mid-to-deep blocks, where the attention maps are already more spatially local. Therefore, feature conflicts caused by early attention broadcasting between different LoRAs are limited in magnitude. In the revision, we added this analysis and tempered the wording from "mathematically negligible" to an evidence-backed statement that the cross-region LoRA perturbation is empirically small in early globally mixing blocks and becomes dominant only in more local semantic blocks.
>
> **Manuscript location.** Sec. 3.1 includes Fig. 4 and the accompanying discussion on LoRA perturbation norms, and the original "mathematically negligible" phrasing has been replaced by an empirical mechanism statement.

---

> > ### Author Response · Authors · 2026-07-04
> > **Response to Reviewer 4TEv (part2)**
> >
> > ### Critical Concern 3: Runtime Comparison in the Main Body
> >
> > > **Reviewer concern.** The main body does not compare the running time of the proposed method with other methods, especially methods that rely on auxiliary segmentation models. Although this comparison is provided in Appendix F, it should be moved to the main body because it supports the motivation that using intrinsic diffusion information avoids the cost of auxiliary models.
> >
> > **Response.** We thank the reviewer for this helpful suggestion. We agree that the runtime comparison is directly tied to one of the main motivations of FreeFuse and should be visible in the main paper rather than only in the appendix.
> >
> > In the revision, we moved the runtime comparison previously reported only in the appendix into the main experimental section, together with the accompanying discussion. This makes the efficiency tradeoff explicit in the main body: methods using auxiliary segmentation models require an additional model call and preprocessing stage, whereas FreeFuse obtains the routing signal from the diffusion model's intrinsic attention information. The appendix still contains the extended latency analysis, but the key runtime table and its interpretation are now presented in the main text.
> >
> > **Manuscript location.** Sec. 4.1 includes the runtime comparison as Tab. 3. The extended latency and computational-cost analysis remains in Appendix H as Tab. 7.

---

### Review · Reviewer_EP1g · 2026-06-24

**Summary Of Contributions:**

FreeFuse is proposed to mitigate conflicts arising from the simultaneous use of multiple LoRAs in multi-subject image generation. It assigns spatial regions to different subjects and restricts each corresponding LoRA to operate within its assigned region. FreeFuse requires no additional training and achieves competitive character identity preservation. However, its core assumptions are strong, and the validation of the routing mechanism itself remains insufficient.

**Audience:**

Yes

**Audience Explanation:**

Multi-subject generation with multiple LoRAs is a practically relevant problem. The core idea of FreeFuse may be informative for researchers working on personalized diffusion models, modular adaptation, and test-time generation control.

**Broader Impact Concerns:**

No specific concerns.

**Claims And Evidence:**

No

**Claims Explanation:**

The method shows promise and improves character identity preservation. However, the routing strategy is insufficiently specified, and the rigid one-to-one LoRA assignment limits the scope of the method. In addition, the ablation study does not directly validate the contribution of the Router itself, and the paper does not quantitatively evaluate performance as the number of LoRAs increases.

**Requested Changes:**

Critical for acceptance:

1. The method relies on a strong one-subject-to-one-LoRA assumption. This rigid mapping does not address settings where a single subject is jointly described by multiple adapters, particularly in competing or semantically overlapping boundary regions.
2. The implementation scope of the Router is insufficiently specified. It is unclear which LoRA target modules are gated, which layers are covered by the LoRAs, and during which denoising steps routing is applied.
3. The ablation study is incomplete. The current results do not directly isolate or validate the contribution of the Router itself.
4. Although the paper aims to mitigate multi-LoRA conflicts, it lacks quantitative quality evaluations as the number of LoRAs increases. Consequently, it is difficult to assess whether the method scales to higher-conflict settings.

Would strengthen the work:

The claimed intrinsic segmentation capability is foundational to the method, yet it is supported only by relatively limited samples and a simple analysis. More extensive analysis is needed to establish its robustness and generality.

---

> ### Author Response · Authors · 2026-07-04
> **Response to Reviewer EP1g (part1)**
>
> ### Critical Concern 1: Multi-Adapter Routing for a Single Subject
>
> > **Reviewer concern.** The method relies on a strong one-subject-to-one-LoRA assumption. This rigid mapping does not address settings where a single subject is jointly described by multiple adapters, particularly in competing or semantically overlapping boundary regions.
>
> **Response.** We thank the reviewer for pointing out that our previous description did not state clearly enough how LoRA adapters are associated with subjects. FreeFuse does **not** assume a strict one-subject-to-one-LoRA mapping. Instead, it supports settings where multiple LoRAs jointly describe the same subject. For example, one subject can be described by a Lady Gaga identity LoRA together with a Christmas-hat LoRA and a pink-gradient Korean-style dress LoRA. Another example is a Harry Potter identity LoRA combined with a black-and-white line-art style LoRA. These multi-adapter compositions are already shown in our qualitative examples, and the revised manuscript now makes this capability explicit.
>
> **Manuscript location.** Sec. 3.1, "Masking LoRA Outputs for Effective Subject Feature Preservation", now introduces semantic groups $G\_s$ and adapter sets $\\mathcal{A}\_s$ before Eq. (1). Sec. 3.3 further explains how shared style adapters are assigned to the relevant group-level adapter sets.
>
> FreeFuse supports two ways for multiple LoRAs to jointly affect one subject. First, for localized attribute adapters such as "wearing a Christmas hat", the attribute LoRA can compete for and occupy the corresponding subregion of the character. In this case, the top part of the character's head is routed to the Christmas-hat LoRA, which adds the hat while the identity LoRA remains active on the rest of the character region. Second, for subject-level or style-level composition, such as a black-and-white line-art Harry Potter, we can assign the style LoRA and the identity LoRA to share the same subject mask. The two adapters then act together on the same routed region, producing the target identity while changing its visual style.
>
> To make this relationship precise, we revised Eq. (1) into a group-level form:
>
> $$h^{\\prime}\_p = h\_p + \\sum\_{s=1}^{S}\\mathbb{I}(p \\in R\_s)\\sum\_{a \\in \\mathcal{A}\_s}\\Delta\\theta\_a(x\_p).$$
>
> Here $R\_s$ denotes the semantic region of subject/group $s$, and $\\mathcal{A}\_s$ denotes one or more LoRA adapters assigned to this region. Under this formulation, adapters assigned to the same group are accumulated within their shared region. A style or attribute LoRA can also be included in $\\mathcal{A}\_s$ when it is intended to modify subject $s$, such as assigning the black-and-white line-art LoRA and the Harry Potter identity LoRA to the same subject group. The exclusivity constraint is therefore applied only between competing subject groups. This clearer description avoids the misleading impression that FreeFuse is limited to one adapter per subject.

---

> > ### Author Response · Authors · 2026-07-04
> > **Response to Reviewer EP1g (part2)**
> >
> > ### Critical Concern 2: Router Implementation Scope
> >
> > > **Reviewer concern.** The implementation scope of the Router is insufficiently specified. It is unclear which LoRA target modules are gated, which layers are covered by the LoRAs, and during which denoising steps routing is applied.
> >
> > **Response.** We thank the reviewer for pointing out that the implementation scope should be stated more explicitly. We have added the following details to the revised manuscript so that the Router configuration is fully reproducible.
> >
> > | Item | Specification used in our experiments |
> > |---|---|
> > | Gated LoRA modules | The Router gates the LoRA residuals in the denoising transformer. For double-stream FLUX blocks, spatial image-token routing is applied to `to_q`, `to_k`, `to_v`, `to_out`, and the image feed-forward branch `ff`. For single-stream FLUX blocks, routing is applied to `to_q`, `to_k`, `to_v`, `proj_mlp`, and `proj_out`. Text-token isolation is applied to the corresponding text-side LoRA residuals, including `add_q_proj`, `add_k_proj`, `add_v_proj`, `to_add_out`, and `ff_context` in double-stream blocks. |
> > | Layer coverage | The Router covers all LoRA-bearing FLUX transformer blocks: 19 double-stream blocks, `transformer_blocks.0--18`, and 38 single-stream blocks, `single_transformer_blocks.0--37`. It does not modify the base model weights; it only masks the additive LoRA residuals. |
> > | Denoising schedule | We use $T=28$ denoising steps by default. FreeFuse first runs a mask-extraction phase for the first $K=5$ steps with LoRA disabled and extracts concept maps from the last double-stream block, `transformer_blocks.18`. The final image is then regenerated from the same initial latent for the full 28 steps, with the routed LoRA residuals enabled at every denoising step. |
> >
> > The revised manuscript now includes this implementation paragraph near the method description. This makes clear that the Router is not a partial or single-layer intervention: the mask is estimated from an early denoising stage, then applied consistently to all LoRA residuals in the routed FLUX transformer during the final generation pass.
> >
> > **Manuscript location.** Sec. 3.3, paragraph "Implementation scope", specifies the gated target modules, the 19 double-stream and 38 single-stream FLUX blocks covered by routing, and the Phase 1/Phase 2 denoising schedule.
> >
> > ### Critical Concern 3: Router Ablation
> >
> > > **Reviewer concern.** The ablation study is incomplete. The current results do not directly isolate or validate the contribution of the Router itself.
> >
> > **Response.** We thank the reviewer for this suggestion. We have added a direct Router ablation to isolate its contribution. In this setting, we remove the Router while keeping the attention-bias mechanism, so the method becomes a **bias-only** variant. This ablation directly tests whether attention bias alone is sufficient or whether spatially gating the LoRA residuals is necessary.
> >
> > | Method | Char DINOv2 ↑ | Char DINOv3 ↑ | Char DreamSim ↓ | Char CLIP-I ↑ | Obj DINOv2 ↑ | Obj DINOv3 ↑ | Obj DreamSim ↓ | Obj CLIP-I ↑ | ArcFace ↑ | LVFace ↑ | CLIP-T ↑ | HPSv2 ↑ | HPSv3 ↑ |
> > |---|---:|---:|---:|---:|---:|---:|---:|---:|---:|---:|---:|---:|---:|
> > | Ours (Cross-Attn) | 0.4271 | 0.4378 | 0.4838 | 0.5919 | 0.6405 | 0.6832 | 0.3867 | 0.7378 | 0.3119 | 0.1975 | 0.2537 | 0.2762 | 6.704 |
> > | Ours (w/o Postprocessing) | 0.4377 | 0.4733 | 0.4303 | 0.6323 | 0.6055 | 0.6570 | 0.4052 | 0.7392 | 0.3259 | 0.1868 | 0.2868 | 0.2846 | 7.727 |
> > | Ours (w/o Attn bias) | 0.4234 | 0.4691 | 0.4186 | 0.6448 | 0.6589 | 0.6939 | 0.3522 | 0.7635 | 0.3491 | 0.2027 | 0.2882 | 0.2865 | 7.971 |
> > | Ours (w/o Router; bias only) | 0.3759 | 0.4138 | 0.5065 | 0.5855 | 0.6495 | 0.7249 | 0.6546 | 0.7477 | 0.2669 | 0.1523 | 0.2678 | 0.2807 | 6.913 |
> > | **Ours (Full)** | **0.4988** | **0.5235** | **0.3753** | **0.6764** | 0.6393 | 0.6804 | **0.3516** | 0.7499 | **0.4275** | **0.2534** | 0.2766 | 0.2857 | **8.279** |
> >
> > The new bias-only row shows that attention bias alone cannot replace the Router: removing the Router substantially reduces character identity preservation (DINOv2 $0.4988 \\rightarrow 0.3759$, DINOv3 $0.5235 \\rightarrow 0.4138$, DreamSim $0.3753 \\rightarrow 0.5065$) and fine-grained face similarity (ArcFace $0.4275 \\rightarrow 0.2669$, LVFace $0.2534 \\rightarrow 0.1523$). The drop is also reflected in the overall aesthetic score (HPSv3 $8.279 \\rightarrow 6.913$). We included this additional row and the corresponding discussion in the revised ablation section to explicitly validate the contribution of the Router.
> >
> > **Manuscript location.** Sec. 4.3, "Ablation Study", includes the bias-only **Ours (w/o Router)** row in Tab. 2 and discusses the resulting identity drop below the table.

---

> > > ### Author Response · Authors · 2026-07-04
> > > **Response to Reviewer EP1g (part3)**
> > >
> > > ### Critical Concern 4: Scaling to More LoRAs
> > >
> > > > **Reviewer concern.** Although the paper aims to mitigate multi-LoRA conflicts, it lacks quantitative quality evaluations as the number of LoRAs increases. Consequently, it is difficult to assess whether the method scales to higher-conflict settings.
> > >
> > > **Response.** We thank the reviewer for suggesting this important scaling evaluation. We added a new LoRA-count ablation that varies the number of simultaneously composed LoRAs from 2 to 5 and compares FreeFuse with the direct baseline under the same evaluation protocol. Each method generates 100 samples for each LoRA count, resulting in 400 samples per method.
> > >
> > > | #LoRAs | Baseline ArcFace ↑ | FreeFuse ArcFace ↑ | Δ | Baseline HPSv2 ↑ | FreeFuse HPSv2 ↑ | Δ |
> > > |---:|---:|---:|---:|---:|---:|---:|
> > > | 2 | 0.4305 | **0.7002** | +0.2697 | 0.2806 | **0.2806** | +0.0000 |
> > > | 3 | 0.2695 | **0.6623** | +0.3928 | 0.2786 | **0.2804** | +0.0019 |
> > > | 4 | 0.2028 | **0.6285** | +0.4257 | 0.2776 | **0.2809** | +0.0033 |
> > > | 5 | 0.1759 | **0.3705** | +0.1946 | 0.2712 | **0.2773** | +0.0061 |
> > > | Avg. | 0.2697 | **0.5904** | +0.3207 | 0.2770 | **0.2798** | +0.0028 |
> > >
> > > **Figure reference for reviewers.** The trend plots are included in the revised manuscript as Appendix E, Fig. 17. The comment keeps the quantitative table above because tables are displayable in comments.
> > >
> > > As expected, identity preservation becomes harder as more LoRAs are introduced and the conflict level increases. Nevertheless, FreeFuse consistently improves ArcFace ID over the baseline at every LoRA count, with gains of $+0.2697$, $+0.3928$, $+0.4257$, and $+0.1946$ for 2, 3, 4, and 5 LoRAs, respectively. The average ArcFace improvement is $+0.3207$. Importantly, this large identity gain does not come from sacrificing image quality: HPSv2 remains comparable to or slightly higher than the baseline, with an average improvement of $+0.0028$. This result directly evaluates the higher-conflict regime and shows that the proposed routing mechanism scales to more simultaneous LoRAs by preserving subject identity while maintaining visual quality.
> > >
> > > **Manuscript location.** Appendix E, "Scaling to More LoRAs", reports this evaluation in Tab. 5 and visualizes the ArcFace/HPSv2 trends in Fig. 17.
> > >
> > > ### Critical Concern 5: Robustness of Intrinsic Segmentation
> > >
> > > > **Reviewer concern.** The claimed intrinsic segmentation capability is foundational to the method, yet it is supported only by relatively limited samples and a simple analysis. More extensive analysis is needed to establish its robustness and generality.
> > >
> > > **Response.** We thank the reviewer for requesting a more explicit and extensive validation of the intrinsic segmentation behavior. We clarified the scale of the existing analysis: our comparison showing that FreeFuseAttn outperforms other attention-based extraction methods was evaluated on 100 generated samples rather than only on qualitative examples.
> > >
> > > In addition, we added a larger 300-sample step/block sweep to justify the spatiotemporal choice used by FreeFuse. For each denoising step and FLUX double-stream block, we compute $\\mathrm{Precision@10\\%}$, defined as the percentage of the top 10% activated tokens that fall inside the corresponding subject region. We described the sample construction details in the revised manuscript.
> > >
> > > **Figure reference for reviewers.** The step/block heatmap is included in the revised manuscript as Fig. 6 in Sec. 3.2. The comment text reports the key value, mean Precision@10% = 0.704 at block 18, step index 4.
> > >
> > > The sweep identifies the selected location, block 18 at step index 4, as the best-performing cell with mean Precision@10%=0.704. Since the denoising index is zero-based, step index 4 corresponds to the 5th denoising step used in our method. This additional 300-sample analysis supports that our choice of the last double-stream block and early-to-mid denoising stage is not arbitrary, and we included the heatmap and protocol details in the revised manuscript to make the robustness evidence explicit.
> > >
> > > **Manuscript location.** Sec. 3.2 states the 5th denoising step and `transformer_blocks.18` choice, Fig. 6 reports the 300-sample step/block sweep, and Fig. 3 reports the FreeFuseAttn comparison against other attention-based extraction methods.

---

### Author Response · Authors · 2026-07-04

We thank the reviewers for the detailed feedback. Below we summarize each reviewer’s main concerns and the corresponding changes made in the revised manuscript.

### Reviewer EP1g

- **Concern: FreeFuse may rely on a rigid one-subject-to-one-LoRA assumption.**
  **Revision:** We revised the method formulation to use semantic groups $G\_s$, spatial regions $R\_s$, and adapter sets $\\mathcal{A}\_s$. A single subject/group can now contain multiple LoRAs, such as identity + localized attribute + style. This is reflected in the revised Eq. (1) and the accompanying explanation in Sec. 3.1 and Sec. 3.3.

- **Concern: Router implementation scope was underspecified.**
  **Revision:** We added the exact gated FLUX target modules, layer coverage, and denoising schedule. The Router covers all LoRA-bearing FLUX transformer blocks, applies routing to image-token LoRA residuals, isolates text-token LoRA residuals where needed, and uses a two-phase schedule: mask extraction in the first $K=5$ steps and routed generation over $T=28$ steps. See Sec. 3.3.

- **Concern: Router contribution was not directly isolated.**
  **Revision:** We added a bias-only ablation that removes the Router while keeping attention bias. The resulting drops in character similarity, face similarity, and HPSv3 directly validate the Router’s contribution. See Sec. 4.3, Tab. 2.

- **Concern: Scaling to more LoRAs lacked quantitative evaluation.**
  **Revision:** We added a LoRA-count ablation for 2--5 simultaneous LoRAs. FreeFuse consistently improves ArcFace ID over direct multi-LoRA activation while keeping HPSv2 comparable or slightly higher. See Appendix E, Tab. 5 and Fig. 17.

- **Concern: Intrinsic segmentation robustness needed stronger evidence.**
  **Revision:** We clarified that the FreeFuseAttn comparison was evaluated on 100 generated samples and added a 300-sample step/block sweep. The sweep confirms block 18 at step index 4 as the best mask-extraction location. See Sec. 3.2 and Fig. 6.

### Reviewer 4TEv

- **Concern: Variables in Eq. (1)--(3) were used before definition.**
  **Revision:** We added explicit definitions for $x\_p$, $h\_p$, $h^{\\prime}\_p$, $\\Delta\\theta\_a(x\_p)$, $Q$, $K$, $V$, $A\_{p,q}$, and $V\_q$ before the equations. See Sec. 3.1.

- **Concern: Early blocks show global attention mixing, so the locality argument needed direct support.**
  **Revision:** We added a LoRA perturbation magnitude analysis over all available LoRAShop test LoRAs. The analysis shows early blocks carry much smaller LoRA perturbations, while mid/deep blocks have $1.80\\times$ larger normalized LoRA weights and $2.67\\times$ larger prompt-activated residuals. We also softened the original “mathematically negligible” wording into an evidence-backed statement. See Sec. 3.1 and Fig. 4.

- **Concern: Runtime comparison should appear in the main body.**
  **Revision:** We moved the key runtime comparison from the appendix into the main experimental section and kept the extended latency discussion in Appendix H. See Sec. 4.1, Tab. 3 and Appendix H, Tab. 7.

### Reviewer LCT8

- **Concern: Implementation details around Eq. (7), TopK, anchors, and the morphological kernel were insufficient.**
  **Revision:** We added $\\tau=4000$, the TopK rule $k=\\max(1,\\lfloor0.1N\\rfloor)$, the anchor count $k=409$ for $1024\\times1024$ FLUX generation, all Eq. (7) dimensions, the output mask shape, and the explicit $2\\times2$ morphological kernel. See Sec. 3.2 and Appendix B.

- **Concern: Evaluation should include public dual-subject protocols and additional baselines.**
  **Revision:** We added an evaluation on the public LoRAShop dual-subject protocol because its evaluation LoRAs and prompting protocol are fully accessible. FreeFuse achieves the best identity preservation and text alignment among the compared methods. We also clarified that our main benchmark is harder than LoRAShop’s public protocol. See Appendix F, Tab. 6.

- **Concern: The spatial masking sufficiency claim needed stronger scientific justification.**
  **Revision:** We revised the claim to state that routing blocks direct cross-region injection of LoRA-specific residuals, while base-model attention remains active for global context. We support this with the locality analysis and the new early-block perturbation analysis. See Sec. 3.1, Fig. 4 and Sec. 3.3.

- **Concern: “Global stylistic adjustments” was vague.**
  **Revision:** We now define whole-image stylistic adjustment as the case where a user provides a style LoRA intended to change the whole image style, such as a Van Gogh-style or Miyazaki-style LoRA. Such adapters are assigned to each relevant $\\mathcal{A}\_s$, or to a background/full-canvas region when needed. See Sec. 3.3.